# Vagus nerve stimulation boosts the drive to work for rewards

Monja P. Neuser [1], Vanessa Teckentrup [1], Anne Kühnel [1,2], Manfred Hallschmid [3,4,5], Martin Walter[1,6,7,8] & Nils B. Kroemer [1✉]

Interoceptive feedback transmitted via the vagus nerve plays a vital role in motivation by tuning actions according to physiological needs. Whereas vagus nerve stimulation (VNS) reinforces actions in animals, motivational effects elicited by VNS in humans are still largely elusive. Here, we applied non-invasive transcutaneous auricular VNS (taVNS) on the left or right ear while participants exerted effort to earn rewards using a randomized cross-over design (vs. sham). In line with preclinical studies, acute taVNS enhances invigoration of effort, and stimulation on the left side primarily facilitates invigoration for food rewards. In contrast, we do not find conclusive evidence that acute taVNS affects effort maintenance or wanting ratings. Collectively, our results suggest that taVNS enhances reward-seeking by boosting invigoration, not effort maintenance and that the stimulation side affects generalization beyond food reward. Thus, taVNS may enhance the pursuit of prospective rewards which may pave avenues to treat motivational deficiencies.

[1] Department of Psychiatry and Psychotherapy, University of Tübingen, Calwerstraße 14, 72076 Tübingen, Germany. [2] Department of Translational Research in Psychiatry, Max Planck Institute of Psychiatry and International Max Planck Research School for Translational Psychiatry (IMPRS-TP), Kraeplinstraße 2-10, 80804 Munich, Germany. [3] Department of Medical Psychology and Behavioral Neurobiology, University of Tübingen, Otfried-Müller-Straße 25, 72076 Tübingen, Germany. [4] German Center for Diabetes Research (DZD), Otfried-Müller-Straße 10, 72076 Tübingen, Germany. [5] Institute for Diabetes Research and Metabolic Diseases of the Helmholtz Center Munich at the Eberhard Karls University Tübingen, Otfried-Müller-Str. 10, 72076 Tübingen, Germany. [6] Department of Psychiatry and Psychotherapy, Otto von Guericke University Magdeburg, Leipziger Straße 44, 39120 Magdeburg, Germany. [7] Department of Psychiatry and Psychotherapy, University Hospital Jena, Philosophenweg 3, 07743 Jena, Germany. [8] Department of Behavioral Neurology, Leibniz Institute for Neurobiology, Brenneckestr 6, 39118 Magdeburg, Germany. ✉email: nils.kroemer@uni-tuebingen.de

In life, pursuing rewards often comes at a cost which is epitomized in the idiom that there is no free lunch. Imagine the cafeteria at work serves decent food, but there is also a stellar restaurant offering your favorite dish as an affordable lunch special. Although the prospective benefits are different, we may go for the cafeteria instead of the restaurant because it is close by. In such cases, we are confronted with the challenge to integrate the costs of action such as the effort of walking a distance with its anticipated benefits, such as eating a better meal. According to economic theories, an optimal decision-maker discounts prospective benefits by the costs of actions incurred[1]. Alternatively, idioms in German and English suggest a second route: you may go with your gut in deciding which option to pick and how much effort to put in[2]. To date, these two decision-making strategies have often been portrayed as (more or less) independent processes and, specifically, the role of the gut has been commonly dismissed as primarily figurative[2]. However, there is emerging evidence from preclinical studies pointing to a vital role of gut-derived signals in the regulation of motivation via dopaminergic circuits[3,4]. Although these results challenge the assumption that the gut plays only a figurative role in human motivation, a conclusive experimental demonstration of such a modulation in humans is lacking to date.

To ensure body homeostasis, it is pivotal to regulate motivation and energy metabolism in concert. This process is called allostasis[5]. As an important part of the autonomic nervous system, the vagus nerve is involved in allostatic regulation through its afferent and efferent pathways[6]. To control food intake, vagal afferents primarily provide negative feedback signals[7], routed via the nucleus tractus solitarii (NTS). These vagal afferent projections are sufficient as decerebrated rats still terminate meal intake[8]. In line with this idea, chronic vagus nerve stimulation (VNS) has been consistently shown to reduce body weight in animals and humans. Preclinical studies indicate that this is primarily due to reduced food intake[7,9]. Likewise, two recent studies have shown that acute taVNS reduces gastric myoelectric frequency of the stomach[10]. At the same time, acute VNS has reinforcing properties leading to sustained self-stimulation and conditioning preferences for flavors or places via a dopaminergic mechanism[3,11]. Furthermore, activation of vagal afferents regulates learning and memory in rats and humans suggesting a role in reward seeking[12,13]. Therefore, chronic reductions in food intake could be linked to acute increases in motivational drive by a combination of afferent and efferent effects.

Within the feeding circuit, the NTS serves as a hub relaying metabolic information to the midbrain and forebrain[8,14] including to dopaminergic neurons in the substantia nigra[3]. Vagal afferent activation can thereby indirectly modulate key brain circuits involved in reward[15] and energy homeostasis[8], because the presence of nutrients in the gut evokes dopamine release in the dorsal striatum tracking caloric load[4,16]. Notably, the dorsal striatum is known to play a critical role in the allocation of response vigor[17,18], and the invigoration (or energization) of behavior via dopamine signaling[19,20], pointing to a link between energy metabolism and goal-directed action. In addition, noradrenergic signaling has also been shown to facilitate invigoration in monkeys[21]. Such an invigorating mechanism may help to explain why VNS has elicited antidepressive effects, even in patients who were treated for epilepsy and did not show improvement of epileptic symptoms[6]. Taken together, the vagal afferent projections to the NTS are a promising candidate for modulatory input onto brain circuits encoding motivation.

Despite the growing evidence for vagal regulation of goal-directed behavior, it is still unclear whether preclinical findings using predominantly food as reward and invasive stimulation will extend to humans and secondary reinforcers such as money.

Moreover, it is not known whether there is a lateralization of vagal afferent signals in humans, as observed in rodents[3]. Until recently, research on vagal input in humans was limited due to the invasive nature of implanted VNS devices. Today, non-invasive transcutaneous auricular VNS (taVNS) has become a promising avenue for research and, potentially, treatment of various disorders. Commonly, taVNS is applied via the ear targeting the auricular branch of the vagus nerve, where the stimulation elicits far-field potentials[22]. Successful activation of the NTS has been demonstrated in animals after taVNS[23]. Likewise, human neuroimaging studies using fMRI have shown enhanced activity in the NTS and other brain regions related to motivation including the dopaminergic midbrain and striatum after concurrent taVNS[24–26]. Compared to implanted VNS, similar therapeutic effects have been reported after taVNS[27–29]. In line with the hypothesized potential of VNS to alter motivational processes, we recently found that taVNS affects value-based learning in a go/no-go reinforcement learning task[13]. Thus, non-invasive taVNS may provide an effective means to study the endogenous regulation of motivation according to homeostatic needs.

Taken together, the vagus nerve may provide an important interface connecting metabolic signals from the periphery with central nervous circuits involved in goal-directed, allostatic behavior. Here, we tested whether non-invasive taVNS—applied to emulate interoceptive feedback signals—would modulate effort if different rewards are at stake (food or money). To better understand potential changes in motivation, we focus on the motivational phases of invigoration versus effort maintenance. In our task, invigoration relates to how quickly a participant energizes effortful behavior, whereas maintenance relates to how durably effort is kept up[18]. Due to the modulatory effects of taVNS on the brain and on behavior, we hypothesized that taVNS would enhance the invigoration of effort by altering the perceived benefit of effortful behavior, which has been linked to dopamine tone before[30,31]. Similarly, taVNS-induced increases in noradrenaline would also lead to an enhanced invigoration of effort[21]. We also assess whether taVNS alters effort maintenance by reducing the costs of actions, which may point to a serotonergic mechanism instead[32,33]. Moreover, we test whether taVNS applied to the right versus the left ear would generalize beyond the regulation of food reward as suggested by Han et al.[3]. We find that taVNS increases invigoration, but not maintenance of effort or rated wanting, and that the side of the stimulation affects generalization beyond food reward.

## Results

**Invigoration is primarily linked to benefits.** Since we used frequency of repeated button presses instead of grip force in our effort task[34], we first validated the primary outcomes: invigoration and effort maintenance. To this end, we used mixed-effects models predicting either invigoration slopes or average relative frequency of button presses (as indication of maintenance) using the factors reward type (food vs. money), reward magnitude (low vs. high), difficulty (easy vs. hard), and the interaction between reward magnitude × difficulty (Fig. 1). To account for the stimulation effect, we also included stimulation condition (taVNS vs. sham) as well as interactions of stimulation with the other predictors to the model and controlled for order of stimulation conditions and stimulation side at the participant level (see "Methods" section).

In line with economically optimal behavior, participants were quicker to invigorate behavior when more reward was at stake, $b = 5.79$, $t(78) = 4.69$, $p < 0.001$ (Fig. 2a). Higher difficulty reduced invigoration, $b = -2.44$, $t(78) = -3.26$, $p = 0.002$. Within our task, the effect of costs (indexed by difficulty) on

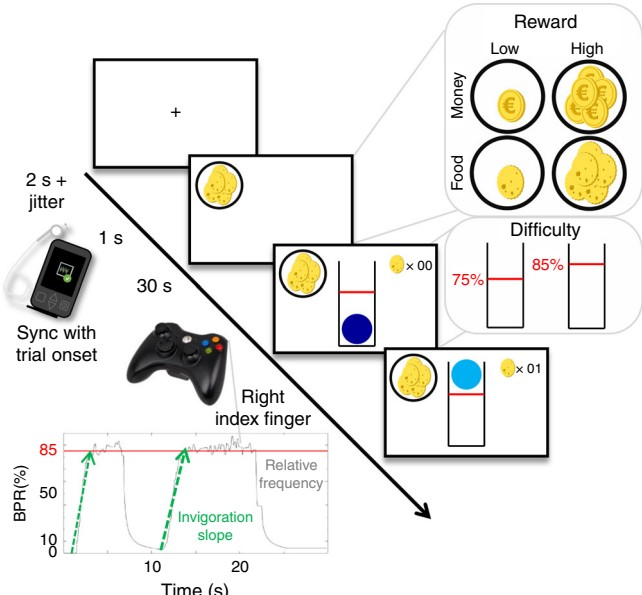

**Fig. 1 Schematic summary of the effort allocation task.** First, a fixation cross is shown. The trial starts in sync with the stimulation and the reward cue is shown for 1 s. During the effort phase, participants have to keep a ball above the red line by vigorously pressing a button with their right index finger to earn rewards. As task conditions, we manipulated reward type (food vs. money), reward magnitude (low vs. high), and difficulty (easy vs. hard). The inset shows a representative time series in one high-difficulty trial depicting effort output as button press rate, BPR, in % relative to the maximum frequency of the participant. Invigoration slopes were estimated to capture how quickly participants ramp up their effort during a trial to obtain the reward at stake. Effort maintenance was estimated by taking the average relative frequency on the trial.

invigoration was only half compared to the effect of benefits (indexed by reward magnitude). Likewise, invigoration was only associated with wanting ratings, $t(78) = 6.14$, $p < 0.001$, but not exertion ratings, $t(78) = 0.15$, $p = 0.88$ (Fig. 2e). Of note, exertion ratings on the preceding trial also did not predict invigoration in the following trial ($b = 0.017$, $p = 0.414$).

Analogous to invigoration, participants showed higher effort maintenance when more reward was at stake, $b = 9.18$, $t(78) = 7.09$, $p < 0.001$ (Fig. 2b). Again, when rewards became more difficult to obtain, effort dropped significantly, $b = -6.71$, $t(78) = -6.66$, $p < 0.001$. Within our task, the effect of costs on effort maintenance was therefore about three quarters compared to the effect of benefits and larger compared to the effect of costs on invigoration. Participants also worked more selectively for large rewards when difficulty was high, leading to a reward magnitude × difficulty interaction, $b = 2.08$, $t(78) = 3.88$, $p < 0.001$. Moreover, effort maintenance was associated with ratings of wanting and exertion, $ts > 8.08$, $ps < 0.001$ (Fig. 2e) and exertion on the previous trial predicted effort maintenance on the following trial, $b = -0.062$, $p < 0.001$, suggesting that it is sensitive to costs of effort. Critically, food and monetary rewards elicited comparably quick invigoration, $b = -0.63$, $t(78) = -0.77$, $p = 0.45$ (Fig. 2c), and effort maintenance, $b = -0.67$, $t(78) = -0.75$, $p = 0.45$ (Fig. 2d), showing that both rewards were comparable in incentive value (Supplementary Fig. 1).

**taVNS increases invigoration for rewards.** After verifying that invigoration primarily tracks wanting of prospective benefits whereas effort maintenance is more strongly affected by difficulty and reflects exertion, we assessed the effects of taVNS (vs. sham)

on the two primary motivational outcomes. In general, participants were faster to invigorate actions during taVNS versus sham (stimulation main effect, $b = 2.93$, 95% CI [0.98, 4.88], $t(78) = 2.943$, $p = 0.004$; Fig. 3; Supplementary Table 1). The increase in invigoration was 5.30% relative to the intercept (55.32). This was more than half of the effect elicited by the 10-fold increase in reward magnitude (i.e., a 10.45% increase relative to the intercept). The corresponding Bayes factor for the main effect of taVNS, $BF_{10} = 7.34$, provided moderate evidence in favor of an increase in invigoration. Furthermore, taVNS-induced effects were stronger for food compared to monetary rewards (stimulation × reward type interaction: $b = 1.33$, $t(78) = 1.998$, $p = 0.049$). Although the side of the stimulation did not affect the main effect of taVNS ($p = 0.947$), taVNS on the left side led to a significantly stronger interaction effect (cross-level interaction of stimulation side on stimulation × reward type, $b = -2.82$, $t(78) = -2.122$, $p = 0.037$). The corresponding Bayes factor did not reach a moderate evidence level, $BF_{10} = 2.40$. Nevertheless, restricting the analysis of the stimulation × reward type effect to the left side of taVNS provided strong evidence for a food-specific effect, $t(39) = 3.172$, $p = 0.003$, $BF_{10} = 11.80$. In contrast, stimulation on the right side did not lead to a stimulation × reward type effect, $t(38) = -0.118$, $p = 0.91$, $BF_{10} = 0.17$ and provided moderate evidence against an interaction. Taken together, stronger taVNS-induced effects for food versus monetary rewards were primarily due to a food-specific increase after stimulation on the left side.

Conversely, taVNS did not significantly enhance effort maintenance compared to sham stimulation ($b = 1.21$, $t(78) = 1.715$, $p = 0.090$, $BF_{10} = 0.51$; Supplementary Table 2), and differences between conditions were not stronger for food rewards ($p = 0.86$) or modulated by the side of the stimulation ($ps > 0.20$; for individual estimates of stimulation effects, see Fig. 4). Crucially, taVNS-induced increases in invigoration were also significant if trial-based effort maintenance was controlled for in the mixed-effects model ($b_{adj} = 1.99$, $p = 0.025$). Moreover, we observed no taVNS effects on the duration of work segments ($p = 0.17$, Supplementary Fig. 2) and the temporal allocation of effort was comparable within and across trials (Supplementary Fig. 3). Thus, our results suggest that taVNS primarily boosts invigoration without altering effort maintenance, although more data would be needed to conclusively demonstrate the absence or presence of a small taVNS effect on maintenance.

**taVNS boosts the drive to work for less-wanted rewards.** Increases in invigoration during taVNS suggest an increase in the prospective benefit of obtaining rewards, but several potential mechanisms may account for the reported changes. One possibility is that taVNS increases subjectively rated wanting of rewards (i.e., perceived benefits of instrumental action). However, the absence of a stimulation main effect, $t(78) = 0.488$, $p = 0.63$, or a stimulation × reward type interaction in predicting wanting ratings, $t(78) = -0.341$, $p = 0.73$, argues against this explanation. Another possibility is that taVNS decreases subjectively rated exertion after working for rewards (i.e., perceived costs of instrumental action), but this was also not the case (stimulation main effect, $t(78) = 0.704$, $p = 0.48$). Absence of taVNS-induced changes in rated wanting and exertion thus point to a difference in the drive to work for rewards.

To test for a potential change in an effort utility slope (i.e., changes in effort per one-unit difference in wanting), we estimated the correspondence of wanting ratings and invigoration for each condition (stimulation and reward type) at the group level using robust regression (see "Methods" section; Fig. 5a). Put simply, the effort utility slope captures how valuable the reward must be to pay for the effortful invigoration, and lower utility slopes indicate that participants invest comparatively more in

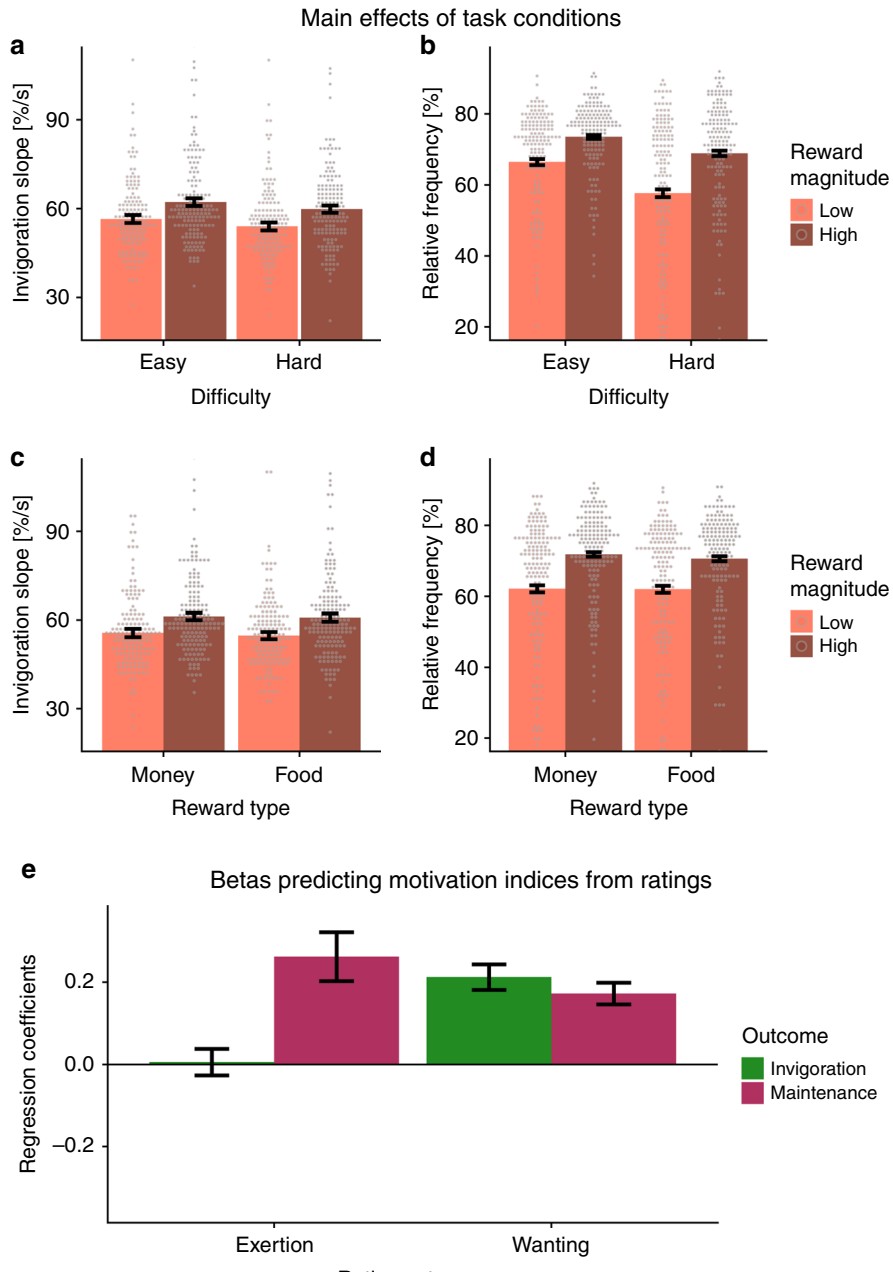

**Fig. 2 Invigoration is associated with reward magnitude and wanting, but not with exertion. a** Participants were quicker to invigorate if more reward was at stake, $p < 0.001$, and slower if difficulty was high, $p = 0.002$. **b** Participants exerted more effort when more reward was at stake, $p < 0.001$, and less when it became more difficult to obtain it, $p < 0.001$. Moreover, they worked more for difficult reward when the magnitude was high, $p < 0.001$. **c** Food and monetary rewards elicited comparable invigoration, $p = 0.45$. **d** Food and monetary rewards elicited similar investment of effort, $p = 0.45$. **e** Effort maintenance was related to both ratings of exertion and wanting, $ps < 0.001$, but invigoration was only related to wanting, $p < 0.001$, not exertion, $p = 0.88$. $n = 7776$ trials of 81 participants; dots depict condition means per participant, bars depict data as sample mean values; error bars depict 95% confidence intervals at the trial level (**a–d**) or of fitted coefficients at the participant level (**e**). %/s = button press rate in % of maximum per s. Statistics refer to two-sided t-contrasts of the mixed-effects models reported in Supplementary Tables 1, 2, approximate degrees of freedom = 78, no adjustments for multiple comparisons. Source data are provided as a Source Data file.

light of decreasing returns. Crucially, we observed significantly reduced effort utility slope coefficients after taVNS (stimulation main effect: $p_{perm} = 0.031$) except for monetary reward after taVNS on the left side (Fig. 5a, b). Collectively, these results suggest that taVNS induced faster invigoration of rewards at stake as if they conferred a higher incentive value. This supports the interpretation that taVNS may bias the utility of instrumental action via an increase of its prospective benefit.

**taVNS does not alter cost-evidence accumulation.** To evaluate whether taVNS alters the integration of interoceptive signals tracking cost evidence, we used a previously established computational model of effort allocation[35]. In the model (Fig. 6a), decisions to stop or resume effort are guided by cost evidence. The signal reflecting cost evidence is accumulated during exertion of effort until it reaches an upper bound. In contrast, taking a break dissipates cost evidence until a lower bound is reached

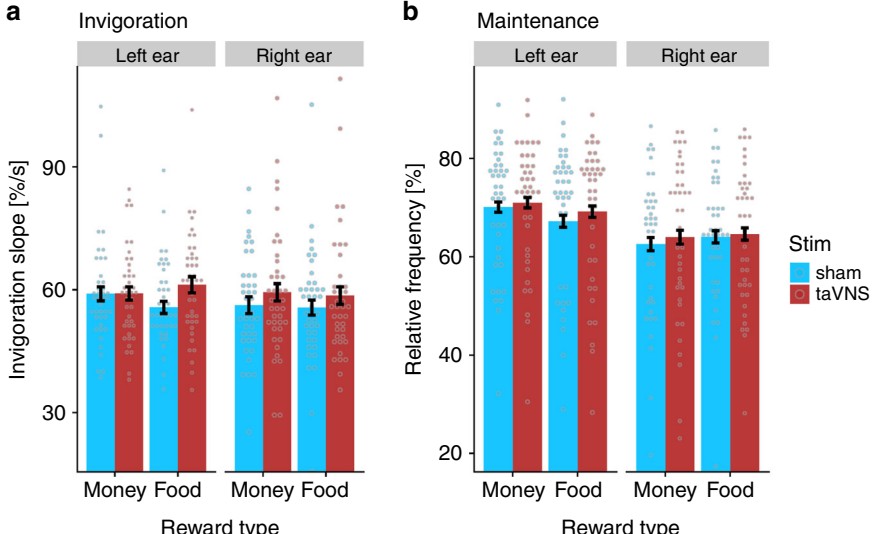

**Fig. 3 Transcutaneous auricular vagus nerve stimulation (taVNS) increases invigoration. a** During taVNS, participants were faster to invigorate instrumental behavior (stimulation main effect, $p = 0.004$, $BF_{10} = 7.34$). The invigorating effect of taVNS was significantly more pronounced for food vs. monetary rewards (stimulation × reward type, $p = 0.049$) which was primarily driven by stimulation side (cross-level interaction, $p = 0.037$). **b** In contrast to invigoration, taVNS did not enhance effort maintenance compared to sham, $p = 0.09$, $BF_{10} = 0.51$. $n = 7776$ trials of 81 participants; dots depict condition means per participant; bars depict data as sample mean values; error bars depict 95% confidence intervals at the trial level. %/s = button press rate in % per s. Note that apparent differences between sham conditions were not significant due to considerable inter-individual differences ($ps > 0.072$). Statistics refer to two-sided t-contrasts of the mixed-effects models reported in Supplementary Tables 1, 2, approximate degrees of freedom = 78, no adjustments for multiple comparisons. Bayes factors were calculated using Bayesian two-sided t-tests based on order-corrected ordinary least squares effects of individual estimates. Source data are provided as a Source Data file.

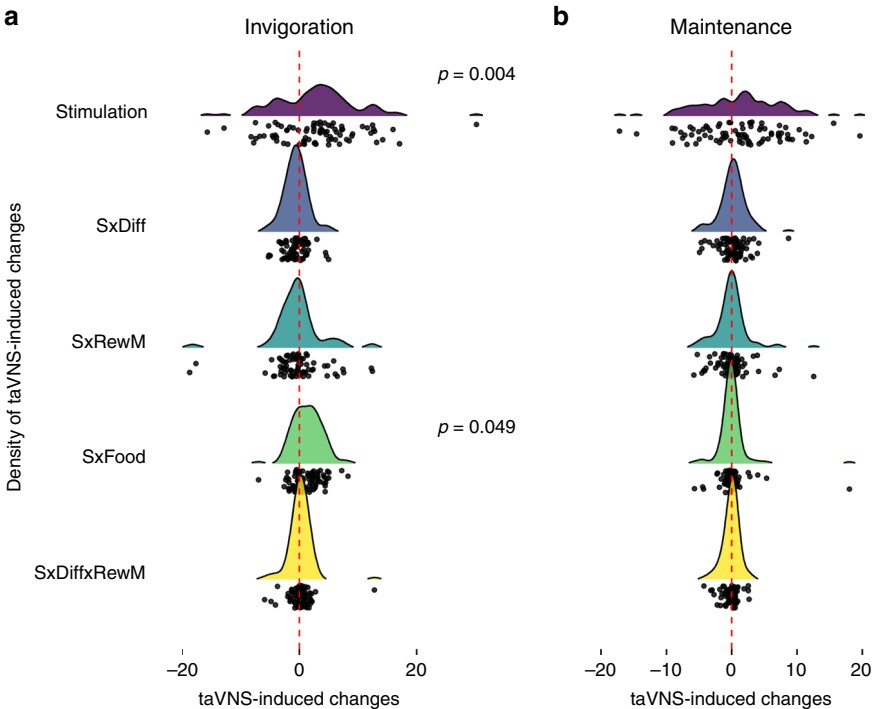

**Fig. 4 Individual estimates and group densities of the stimulation effects induced by transcutaneous auricular vagus nerve stimulation (taVNS). a** During taVNS, participants showed an increase in invigoration across conditions (main effect of stimulation, S; $b = 2.93$, 95% CI [0.98, 4.88], $p = 0.004$). The S × Food interaction was significantly higher during taVNS, ($b = 1.33$, 95% CI [0.03, 2.63], $p = 0.049$) which was primarily driven by stimulation on the left side. **b** No significant changes in effort maintenance were induced by taVNS. Diff = difficulty, RewM = reward magnitude. The plot depicts empirical Bayes estimates (after taking the distribution across the group into account). Statistics refer to two-sided t-contrasts of the mixed-effects models reported in Supplementary Tables 1, 2, approximate degrees of freedom = 78, no adjustments for multiple comparisons. Source data are provided as a Source Data file.

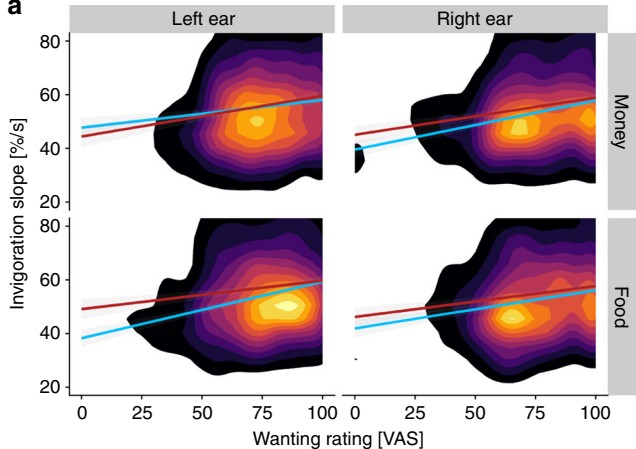

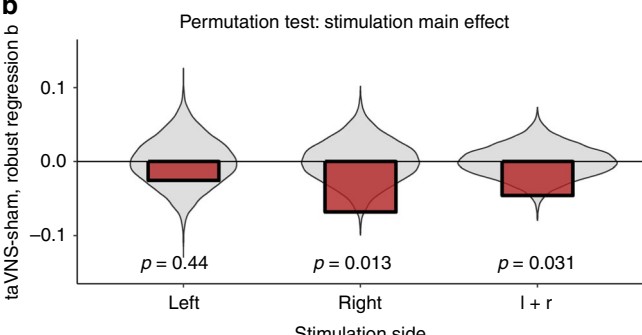

**Fig. 5 Vagus nerve stimulation boosts the drive to work for less wanted rewards. a** Overall, participants are slower to invigorate behavior if they want the reward at stake less as depicted in the 2d-density polygon (brighter colors indicate higher density of data). In line with univariate analyses, robust regression lines show that the slope reflecting the association between invigoration and wanting is decreased by transcutaneous auricular vagus nerve stimulation (taVNS: red line; sham is depicted as blue line). Again, no change was observed for monetary rewards after taVNS on the left side. **b** Compared to permuted data, taVNS induces significant changes in the association between invigoration and wanting. By fitting robust regression coefficients, $b$, after permuting the labels for taVNS vs. sham stimulation, we compared the observed difference in slopes for taVNS—sham (in red) to a null distribution (violin plot in gray). This two-sided permutation test showed a significant main effect across both stimulation sides (l + r = left + right) and for taVNS on the right, but not the left side. On the left side, we observed an interaction with reward type instead, $p = 0.029$. VAS = visual analog scale. $p$-values are unadjusted for multiple comparisons. Source data are provided as a Source Data file.

where the participant resumes effort. To estimate potential changes induced by taVNS, we used a hierarchical Bayesian estimation of a previously established model (see "Methods" section). Critically, there was little indication of changes in cost-evidence accumulation induced by taVNS, as all posterior densities of the taVNS parameters contained 0 in their credible interval. Thus, $BF_{10}$ were between 0.002 and 0.050 providing strong support for the null hypothesis (Fig. 6b, Supplementary Table 3). Notably, individual parameter estimates successfully recovered the individual mean work and rest segment length for all conditions (Fig. 6c).

## Discussion

Although the vagus nerve is known to play a vital role in the regulation of food reward-seeking[3,12], the modulatory effects of

vagal afferent signals on human motivation were largely elusive to date. Here, using non-invasive taVNS, we demonstrate that stimulation of the vagus nerve increases the invigoration to work for rewards in humans. Moreover, we show that the side of the stimulation affects the generalization of the invigorating effect of taVNS. In line with preclinical studies[3], taVNS on the left side affects invigoration primarily when food rewards, but not monetary rewards, are at stake. However, taVNS does not significantly increase effort maintenance or alter rated wanting and exertion during the task. Instead, taVNS increases the drive to work for rewards, particularly when they are wanted less, suggesting a boost in the utility of effort. These motivational effects are well in line with the hypothesized taVNS-induced increase in dopamine tone[13,30,31], although this link needs to be directly investigated in future research. Our results shed light on the role of peripheral physiological signals in regulating instrumental behavior[3,4,16,36,37] and highlight the potential for non-invasive brain stimulation techniques to improve aberrant reward function.

Reward seeking within our task can be dissociated into two key facets: invigoration and effort maintenance. Whereas taVNS does not increase maintenance, it improves invigoration of physical effort. Invigoration has been conclusively linked to dopaminergic transmission in animals[20,30,31] and humans[20,38–40] before. The associations of invigoration slopes with reward magnitude and rated wanting, but not rated exhaustion in our task support the interpretation that the speed of invigoration is primarily related to the prospective benefit of actions and largely independent of the costs incurred by effort. These effects cannot be explained by a nonmotivational motor effect. First, there is good evidence that invigoration is dependent on reward magnitude[17] and we also report a quicker invigoration when large rewards are at stake in our task. Second, invigoration is associated with wanting ratings, but not ratings of exertion on the same trial (Fig. 2e) or exertion on the previous trial. Relatedly, motivational aspects of motor control contribute to motor symptoms in Parkinson's disease[41], which is modulated by alterations in dopaminergic transmission[42]. Thus, one plausible explanation is that the hypothesized taVNS-induced increase in dopamine tone acts comparable to an increase in the average rate of rewards[31,43,44]. Such an increase in the assumed reward rate would make leisure more costly because an agent is missing out on potential benefits, thereby facilitating the rapid approach of prospective rewards[44]. Likewise, a dopamine-induced boost in the expected value of effort would also increase invigoration[40] and could lead to the observed change in the effort utility slope. Alternatively, the facilitating effect of taVNS on invigoration could also be due to a modulatory effect on the noradrenergic system. VNS increases activation of the locus coeruleus[24] and facilitates noradrenergic transmission[45], which is known to affect energization of effort[21]. However, noradrenergic effects are not independent of potential dopaminergic effects[46] and electrical stimulation of the locus coeruleus elicits a co-release of noradrenaline and dopamine in rats[47]. Thus, although changes in monoaminergic transmission induced by VNS could explain increases in invigoration, more research is needed to dissociate dopaminergic and noradrenergic effects. Notwithstanding, these hypothesized mechanisms would be well in line with the previously reported modulatory input of the vagus nerve and the NTS in reward-seeking behavior[3,8,12]. Furthermore, taVNS-induced effects on invigoration could also be partly driven by an alteration of bodily precision[48] that is by modulating the sensitivity to interoceptive signals that guide motivation[49] and decision making[50]. Taken together, these findings support the interpretation that vagal afferents play an important role in tuning instrumental actions in humans according to interoceptive feedback.

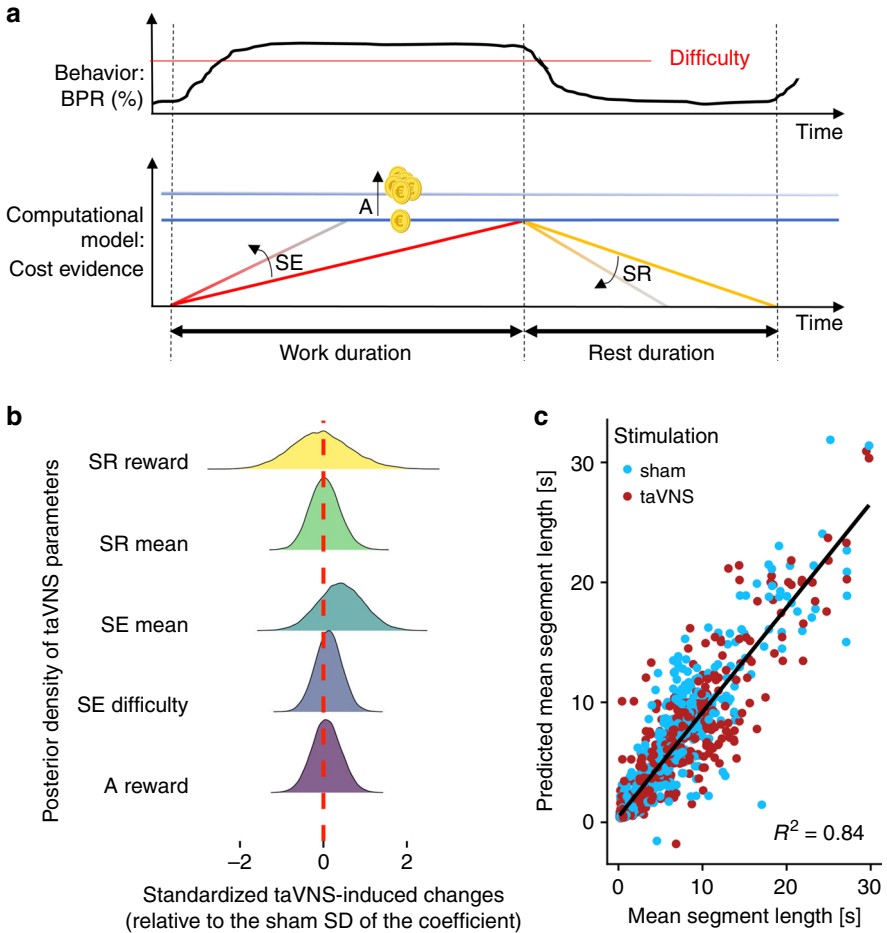

**Fig. 6 Vagus nerve stimulation does not affect cost-evidence accumulation. a** Schematic summary of the computational model of effort allocation as proposed by Meyniel et al.[34]. **b** Posterior densities of transcutaneous auricular vagus nerve stimulation (taVNS)-induced changes in the model parameters. The credible intervals of the posteriors all include 0 suggesting that there is no significant effect on cost-evidence accumulation. **c** Simulations show that the length of segments can be recovered very well by the fitted model for both stimulation conditions. Source data for **b**, **c** are provided as a Source Data file.

Notably, we observe no taVNS-induced changes in the perceived costs of action or rated exertion. In light of previous results suggesting that taVNS might reduce pain[51] and that cost evidence is accumulated in regions related to pain processing[34], it is conceivable that taVNS could act via an encoding of costs. However, our study provides evidence against such a modulatory role in physically effortful behavioral control. Nevertheless, according to our Bayesian analysis, the taVNS main effect on effort maintenance provides only anecdotal evidence for the null hypothesis and more data will be needed to conclusively support the null or provide further evidence for a potentially true, yet small effect. A functional dissociation of taVNS-induced effects is clinically relevant because cost-evidence accumulation is affected by escitalopram, a selective serotonin reuptake inhibitor and common antidepressant drug[33]. Thus, anti-depressive effects of VNS[27–29,52,53] may act via a different neurobehavioral mechanism on the utility of effort than the commonly used antidepressant, pointing to the potential of complementing currently used pharmacological treatment regimens[54]. Notwithstanding, this hypothesis calls for future research in patients suffering from deficits in invigorating goal-directed behavior[55].

Crucially, we show that the generalization of taVNS-induced increases in invigoration is dependent on the side of the stimulation. This observation is well in line with the stronger induction of dopaminergic transmission after stimulation of the right

compared to the left nodos ganglion of the vagus nerve in rodents[3]. The specificity of the invigorating effects of left-sided taVNS for food, but not money, suggests that vagal afferent projections to the NTS may alter diverging parts of the motivational circuit in humans as well. Although there is ample evidence for a common core network encoding reward value, there is also conclusive support for functional specificity[56], particularly regarding primary versus secondary reinforcers[57]. Notably, we also observed acute changes in gastric frequency during taVNS applied on the left side suggesting a link to energy metabolism via digestion[11]. The presence of two lateralized signaling pathways[3] may, therefore, enable the more nuanced regulation of reward-seeking behavior prioritizing the regulation of food-seeking according to metabolic state as transmitted via the vagus nerve[7].

The study has several limitations that will need to be addressed in future research. First, although preclinical data[3] and our behavioral results provide a striking precedent for future research, we did not test directly if the invigoration induced by taVNS is indeed due to increases in dopamine tone as taVNS affects other neurotransmitter systems as well[46,58]. Thus, future research using additional pharmacological manipulations of the dopamine system or positron emission tomography imaging is necessary to test this hypothesis in humans. Second, although we provide evidence that taVNS acts primarily by boosting the prospective benefits of acting, it will require more finely resolved follow-up studies to

unravel the exact mechanism leading to faster invigoration. Concurrent neuroimaging may also provide insights into taVNS-induced changes in neural mechanisms subserving cost-benefit decision-making. Third, we only compared one taVNS stimulation side to sham, but future studies could directly test differences due to lateralization within participants. Fourth, although we selected food versus monetary rewards to assess generalization to secondary reinforcers, it is conceivable that other characteristics of the two rewards such as depletion status could affect taVNS-induced effects on invigoration. To better understand the precise behavioral mechanism, it is advisable to extend the set of reinforcers and conditions such as the status of resource depletion. Fifth, we used a stimulation strength that produced a somatic sensation according to the instructions of the therapeutic use of the taVNS device. It is possible that different stimulation protocols could lead to different effects on motivation.

To summarize, we show that non-invasive taVNS increases invigoration to work for rewards without altering effort maintenance. Furthermore, taVNS alters the correspondence between invigoration and subjective ratings of wanting, effectively increasing participants' drive to approach less-wanted rewards. Collectively, our results indicate that taVNS alters motivation primarily by boosting the prospective benefit of work, not by altering the costs associated with maintaining effort. Moreover, as suggested by preclinical research, stimulation at the left ear exerts stronger effects on invigoration when food rewards are at stake. We conclude that taVNS may provide a promising brain stimulation technique to acutely improve motivational syndromes characterized by a lack of vigor to pursue rewards such as apathy[55,59] or anhedonia[60] as these symptoms might be partly caused by aberrant vagal signaling[61,62]. Notwithstanding, it is unknown to date whether acute improvements can predict sustained therapeutic effects of potential taVNS-based treatments, which remains as a vital question for the translation to clinical settings. Our findings also add to the growing literature demonstrating the crucial role of peripheral interoceptive signals in tuning instrumental behavior according to metabolic needs. Intriguingly, our results corroborate recent theories suggesting that the gut–brain reward pathways may bypass the cognitive regulatory system by directly reinforcing behavior[63]. Ultimately, this perspective may help to better understand the etiology of common motivational symptoms across disorders[55].

## Methods

**Participants**. A total of 85 right-handed individuals participated in the study. Each participant had to complete two sessions: one after stimulation of the cymba conchae (taVNS) and one after a sham stimulation at the earlobe. For the current analysis, four participants had to be excluded ($n = 3$: did not finish the second experimental session, for example due to sick leave, $n = 1$: was assigned an incorrect maximum of button press frequency precluding comparison of the two sessions) leading to a total sample size of $n = 81$. Out of the 81 participants, 41 completed the effort task during left-sided taVNS, whereas 40 completed the effort task during right-sided taVNS. Participants were physically and mentally healthy, German speaking, and right-handed, as determined by a telephone interview (47 women; $M_{age} = 25.3$ years $\pm$ 3.8; $M_{BMI} = 23.0$ kg per m$^2 \pm 2.95$; 17.9–30.9). The study has been approved by the local ethics committee (the institutional review board of the Faculty of Medicine, University of Tübingen) and was conducted in accordance with the ethical code of the World Medical Association (Declaration of Helsinki). All participants provided written informed consent at the beginning of Session 1 and received either monetary compensation (32 € fixed amount) or course credit for their participation. Moreover, they received money and a breakfast (cereal + chocolate bar) depending on their task performance.

**taVNS stimulation device**. To stimulate the auricular branch of the vagus nerve, we used the NEMOS® stimulation device (cerbomed GmbH, Erlangen, Germany). These taVNS devices have been previously used in clinical trials[64,65] and proof-of-principle neuroimaging studies[24]. The stimulation protocol for the NEMOS is preset to a biphasic impulse frequency of 25 Hz with a stimulation duration of 30 s, followed by a 30 s off phase. However, during the effort task, pauses were controlled by the experimenter and shortened to align taVNS with the effort phases on

each trial. The electrical current is transmitted by a titanium electrode placed at the cymba conchae (taVNS) or earlobe (sham) of the ear[24]. To match the subjective experience of the stimulation, intensity was determined for each participant and each condition individually to correspond to mild pricking ($M_{tVNS} = 1.28 \pm 0.58$; 0.2–3.1 mA; sham: $M_{sham} = 1.85 \pm 0.63$; 0.5–3.1 mA). Due to the matching procedure, participants did not guess better than chance, which stimulation condition they had received in each session (recorded guesses: 148, correct guesses: 79, accuracy: 53.4%, $p_{binom} = 0.18$).

**Effort allocation task**. In the effort allocation task, participants had to collect food and money tokens throughout the task by exerting effort (i.e., repeatedly pressing a button with the right index finger). The task was adapted from Meyniel et al.[34] and used frequency of button presses instead of grip force to measure physical effort, analogous to preclinical studies of lever pressing[39,66]. At the end of the session, tokens were exchanged for calories (cereal + chocolate bar as snack) or money at a rate of 1 kcal or 1 cent per 5 tokens.

Every trial started with the presentation of the reward at stake for 1 s. The prospective reward could be either food (indicated by a cookie), or money (indicated by a coin). We varied the magnitude of the prospective reward as 1 symbol signaled a low magnitude (1 point per s) whereas several symbols signaled a high reward magnitude (10 points per s). On average, participants won 362.8 kcal and 3.78 € per session. Next, a tube containing a blue ball appeared on the screen. To earn reward points, participants had to vertically move the ball above a certain difficulty level by repeatedly pressing a button on the controller with the right index finger. Difficulty corresponded to a relative frequency threshold and was indicated by a red line. For every second that the ball was kept above this threshold (indicated by a change of color from dark to light blue), reward points were accumulated and tracked by a counter in the upper right corner of the screen (Fig. 1). Difficulty was varied by alternating the red threshold line between 75 and 85% (counterbalanced order across participants) of the individual maximum frequency. To smooth the movement of the ball for display on screen, we used a moving average algorithm with exponential weighting ($\lambda = 0.6$). Hence, when participants stopped working or reduced the frequency, the ball fell quickly yet slowed down.

After every effort phase of a trial, participants were presented sequentially with two visual analog scales (VAS) inquiring about exertion and wanting of the reward at stake. The task comprised 48 trials. The instructions emphasized that the task was too difficult to always keep the ball above the red line and participants were encouraged to take breaks at their convenience to recover during trials, so that they could try to exceed the threshold again. After half of the task, participants could take a short break to recuperate. After completing the task, participants were shown the total amount of tokens they had collected. Only completed seconds were rewarded in tokens. The task was presented using Psychophysics toolbox v3[67,68] in MATLAB v2017a.

**Experimental procedure**. Experimental sessions were conducted in a randomized, single-blind crossover design. Participants were required to fast overnight (>8 h prior to the visit) and sessions started between 7:00 am and 10:15 am lasting about 2.5 h each. In the beginning of the first session, participants provided written informed consent. Participants selected their favorite type of cereal out of four options (dried fruits, chocolate, cookies, or honey nut; Peter Kölln GmbH & Co. KGaA, Elmshorn, Germany). They were instructed that they would collect energy and money points depending on their task performance later. Earned energy points would be converted into the participants' breakfast serving consisting of cereal and milk scaled accordingly. During the experiment, participants could drink water ad libitum.

After a set of state ratings, participants completed a practice of the effort task intended to estimate the maximum frequency of button presses for every individual. During two initial trials of 10 s length each, a tube containing a blue ball appeared on the screen. Participants were encouraged to move the ball upwards within the tube by repeatedly pressing a button on the Xbox 360 controller (Microsoft Corporation, Redmond, WA) with their right (dominant) index finger. By moving the ball, they were also moving a blue tangent line on the vertical axis marking the highest position reached by the ball so far. In contrast to the ball, this line would remain to depict the maximum frequency of button presses achieved so far (peak frequency) even when they stopped pressing the button. Participants were instructed to push the line as high as they could. Next, participants completed a short practice analogous to the task consisting of eight trials. All possible combinations of task difficulty (easy vs. hard), reward magnitude (low vs. high), and reward type (food vs. money) were presented in a randomized order and a short break was included after four trials. Critically, these practice trials were also used to update the maximum frequency if participants exceeded the previous level achieved during training. At the end of the practice, participants also received feedback about the reward they would have won to provide a reference for the following experiment.

After practicing the effort task, the taVNS electrode was placed either on the left or the right ear. In line with the procedure by Frangos, Ellrich, and Komisaruk[24], the position for the taVNS stimulation position was located at the left ($N = 41$) or right cymba conchae ($N = 40$) whereas the sham stimulation was applied to the earlobe of the same ear. The skin was cleaned at the respective location using a

cotton swab and alcoholic disinfectant. Then, electrode pads were carefully wetted with electrode contact fluid and the electrode was applied by the experimenter. Stripes of surgical tape secured the electrode and the cable in place. For every session and stimulation condition, the stimulation strength was individually assessed using pain VAS ratings ("How strong do you perceive pain induced by the stimulation?" ranging from 0, "no sensation", to 10, "strongest sensation imaginable"). Stimulation was initiated at an amplitude of 0.1 mA and increased by the experimenter by 0.1–0.2 mA at a time. Participants rated the sensation after every increment using the Xbox 360 controller joystick until ratings plateaued around the value 5 (corresponding to "mild pricking") and the stimulation remained active at this level. Next, participants completed a food-cue reactivity task (~20 min), before they performed the effort task (~40 min). Moreover, participants completed a reinforcement learning task[13]. Since the default stimulation protocol of the NEMOS taVNS device alternates between 30 s on and off phases, the off phases were manually shortened to correspond to the duration of the VAS ratings between effort phases during the task. Thus, stimulation and trial onset were initiated by the experimenter to commence in sync.

Then, participants had the taVNS electrode removed and received their breakfast and a snack according to the food reward (energy) points earned (for details, see Supplementary Information (SI)). To conclude the first session, all participants received their wins as part of the compensation. Both visits were conducted at approximately the same time within a week, usually within 3–4 days and followed the same standardized protocol. After the second session, participants either received monetary compensation (32 € fixed amount + wins of Session 2) or course credit (+ wins of Session 2) for their participation.

**Motivation indices and mixed-effects models of stimulation effects**. To isolate the two motivational facets of invigoration and maintenance of effort, we segmented the behavioral data into work and rest segments (for details, see Supplementary Methods). Briefly, to capture invigoration of effort, we estimated the slope of the transition between relative frequency of button presses during a rest segment and their initial plateau during the subsequent work segment (MATLAB findpeaks function). Maintenance of effort was operationalized as the average frequency of button presses during a trial capturing how much effort participants produce over time.

Estimates of invigoration and maintenance at the trial level were then entered in a mixed-effects analysis as implemented in hierarchical linear models (HLM)[69]. Since both outcomes were only moderately correlated, $r = 0.286$, 95% CI [0.25, 0.32], we used two univariate mixed-effects models. Nevertheless, to test for specificity of taVNS-induced effects, we also ran univariate mixed-effects models controlling for the second outcome. To evaluate stimulation effects, we predicted either invigoration or effort maintenance as outcomes using the following predictors: stimulation (sham vs. taVNS), reward type (food vs. money), reward magnitude (low vs. high), difficulty (easy vs. hard, all dummy coded), the interaction between reward magnitude × difficulty, as well as interactions of stimulation with all of these terms. At the level of participants, we included stimulation order and stimulation side (both mean centered) to account for potential differences due to order or the side of the stimulation. To account for individual deviations from fixed group effects, intercepts and slopes were modeled as random effects. Using nested model comparisons ($\chi^2$), we also assessed whether sex and BMI should be included as nuisance regressors along stimulation side and order. Put simply, any additional regressor will improve the fit of a model, but this improved fit comes at the cost of increased complexity. To guard against overfitting, we used model comparisons to evaluate whether the additional complexity of an extended model is justified. For effort maintenance, the extended model (containing four participant level-predictors) fit the data slightly, but significantly better ($p = 0.015$). In contrast, for invigoration, the restricted model (containing only two participant-level predictors) was equivalent in fit compared to the extended model and should be preferred because it is parsimonious ($p > 0.50$). Since the extended models (including sex and BMI as additional covariates) led to slightly lower $p$-values of the stimulation effect without changing the conclusions, we report the restricted model for invigoration and maintenance to aid direct comparisons.

To test for stimulation effects on subjective ratings, we predicted ratings of wanting (related to benefits of action) or exertion (related to costs of action) using the same set of predictors as the models predicting invigoration and effort maintenance. Moreover, to assess the specific associations of invigoration and effort maintenance with subjective ratings, we used mixed-effects models as implemented in R (lmerTest) predicting invigoration or effort maintenance as outcomes, respectively, using wanting and exertion ratings as predictors.

**taVNS-induced changes in effort utility and cost evidence**. To assess if taVNS changes the association of subjectively rated wanting and invigoration (i.e., the utility to work for one unit of wanting), we used robust regression analysis. Robust regression is preferable in the presence of heteroscedasticity and outliers as these issues violate the assumptions of ordinary least squares regression[70]. We ran robust regression analyses at the group level because we were primarily interested in the group-level stimulation effect and many participants had a restricted range in wanting ratings leading to uninformative individual slope estimates. To test for significance, we permuted the vector encoding the stimulation condition and

repeated the robust regression fitting procedure 10,000 times (MATLAB robustfit, weight function huber). We then compared the observed difference in slopes for taVNS−sham to the null distribution to calculate $p$-values. The regression equation included an intercept and the order of stimulation as a nuisance factor.

To better understand how we decide to rest, Meyniel and colleagues have previously proposed a cost-evidence accumulation model[33–35]. The model is based on the theory that decisions to stop and resume effort are guided by cost evidence. The signal underlying cost evidence is accumulated until it reaches an upper (exhaustion) or a lower bound (recuperation). Briefly, work and rest durations are formalized as linear functions of a shared cost-evidence amplitude ($A$), a cost-evidence accumulation slope (SE, work duration), and a cost-evidence dissipation slope (SR, rest duration), respectively (for detailed equations, see SI). All three parameters can in principle be modulated by reward magnitude and difficulty. However, to ensure convergence and limit correlation between parameters, we only included a modulation of $A$ and SR by reward magnitude and of SE by difficulty based on previous model comparisons[33–35]. Moreover, to normalize the slope estimates and improve stability, we set $A_{mean}$ to 1 and estimated all other effects relative to this intercept. To capture taVNS effects, we modeled additive effects for all free parameters in trials with active stimulation. Illustratively, we estimated $SE_{mean} = SE_{sham} + \mathbf{Stim}$ (sham = 0 and taVNS = 1) $\times SE_{taVNS}$ (for details, see Supplementary Methods). To estimate the model, we used a hierarchical Bayesian approach simultaneously incorporating individual level parameter estimates and group level parameter distributions by using Markov Chain Monte Carlo sampling as implemented in R and JAGS[71]. The advantage of this estimation approach is that the evidence (or significance) of a parameter value across the group can be directly evaluated using the posterior distribution of the group level parameter means. We assessed taVNS effects using the credible intervals of additive taVNS parameters. Furthermore, analogous to the results of the mixed-effects models, we calculated Bayes factors (BFs) comparing the posterior distribution to a Cauchy prior ($r = 0.707$).

**Statistical threshold and software**. We used a two-tailed $\alpha \leq 0.05$ for the analyses of our primary research questions: (1) Does taVNS modulate invigoration or effort maintenance across conditions (stimulation main effect)? (2) Does taVNS applied to the left side compared to the right side elicit effects that are less generalizable beyond food rewards as suggested by Han et al.?[3] Other potential interaction effects with reward magnitude or difficulty were assessed at a corrected level, because there was no a priori hypothesis about specificity. Mixed-effects analyses were conducted with HLM v7[72] and lmerTest in R[73]. To determine the evidence for the alternative hypothesis (i.e., taVNS facilitates motivational aspects of goal-directed behavior such as invigoration or effort maintenance) provided by our results, we calculated corresponding BFs based on order-corrected individual estimates of all stimulation effects (calculated using ordinary least squares). We used the default Cauchy prior $r = 0.707$ as implemented in JASP v0.9[74] or, for the cost-evidence accumulation model, in R. We also conducted a prior robustness analysis and changes in the prior would not have led to differences in evidential conclusions. Effort data was processed with MATLAB vR2017-2019a and SPSS v24. Results were plotted with R v3.4.0[75].

**Reporting summary**. Further information on research design is available in the Nature Research Reporting Summary linked to this article.

## Data availability
A reporting summary for this article is available as a Supplementary Information file. Trial-based behavioral data that was used to conduct all analyses will be made publicly available on OSF upon publication of the manuscript: [https://osf.io/58r3c/?view_only=5d1ccee7d67b464bb6f40ebe7ebc844b]. No customized code is necessary to analyze the provided data and MATLAB code used to preprocess the data will be provided upon reasonable request. Source data are provided with this paper.

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

## Acknowledgements

We thank Caroline Burrasch, Franziska Müller, Sandra Neubert, Moritz Herkner, Magdalena Ferstl, and Leonie Osthof for help with data acquisition as well as Jennifer Them and Wiebke Ringels for support in analyzing the data. The study was supported by the University of Tübingen, Faculty of Medicine fortune grant #2453-0-0. N.B.K. received support from the Daimler and Benz Foundation, grant 32-04/19 and the Else Kröner-Fresenius Stiftung, grant 2017_A67. M.P.N. received support from the Else Kröner-Fresenius Stiftung, grant 2017_A67. M.H. received support via grants from the German Federal Ministry of Education and Research (BMBF) to the German Center for Diabetes Research (DZD e.V.; 01GI0925).

## Author contributions

N.B.K. was responsible for the study concept and design. M.P.N. coded the paradigm and collected data under supervision by N.B.K., M.P.N., and N.B.K. conceived the methods, processed the data, and performed the analysis. A.K. and N.B.K. implemented the computational model and the simulations. M.P.N. and N.B.K. wrote the manuscript. M.P.N., V.T., A.K., M.H., M.W., and N.B.K. contributed to the interpretation of findings, provided critical revision of the manuscript for important intellectual content and approved the final version for publication.

## Competing interests

The authors declare no competing interests.
