## [Peer Review File · Nature Communications]

Reviewers' comments:

Reviewer #1 (Remarks to the Author):

Neuser and colleagues present a vagal nerve stimulation study in healthy people, aimed at understanding how motivation to exert effort for reward may be modulated by signal from the body. They use a modified version of a previously implemented effort allocation task, examining how people decide when to work and rest in terms of repeated button presses. They offered participants different types of rewards (food/money), at different magnitudes, with different levels of difficulty (button presses) for obtaining those rewards. They applied vagal nerve stimulation to the left ear in one group, right ear in the other and sham in both groups and examined the effect on the maintenance of effortful exertion and the initial invigoration of movements following periods of rest. Overall, they find that stimulation increases the vigor towards rewards. In addition, the authors argue that this effect may be greater for food rewards than monetary and only affects the maintenance of effort not the invigoration.

This is clearly a highly novel study. They have used a sophisticated study design, reasonable sample sizes, and I believe the primary conclusion will be of great interest to a number of different sub-fields of psychology and neuroscience. However, I do have a number of concerns about the study, which come about largely because the authors have drawn many of their inferences from what should be considered inconclusive non-significant effects, which may arise due to some methodological limitations. Many of these concerns could however be alleviated by the authors simply drawing their conclusions more from the significant effect of TVNS, which is still novel and of interest.

Major

1. Effects of TVNS on different types of reward – I can appreciate the authors desire to examine whether the effects of stimulation might enhance the wanting and motivation for primary reward, which could link to the literature on embodiment etc. However, there are several differences in this experiment between these two types of rewards that serve as a significant limitation for trying to make specific claims. For instance, participants were in a partially fasted state, with the aim of incentivising the wanting and seeking of the primary rewards, but there was no comparable approach for the monetary rewards. Thus, an equally plausible alternative interpretation is that vagal nerve stimulation increases motivation for reinforcers when they have been significantly depleted. I can completely understand the rationale for taking this approach and appreciate you cannot similarly make your participants experience poverty. However, I think this limitation impairs the ability to make strong claims about the specificity of the vagal nerve effects on reward types, but is strongly interpreted throughout the manuscript. But I also think that for such a novel study the strong claims on specificity are also not necessary for this study to still be of interest.

2. Specificity of effects in regards to aspects of “effort” or invigoration. A similar issue to point 1 arises when the claims are made in the paper regarding the effects of TVNS on the two measures of effort. Firstly, it is slightly odd, in a way, that they are treated as separate independent measures, because one

measure effects the other within a trial. However, the two measures are not put together in the same analysis i.e. testing whether “invigoration” is modulated by TVNS taking into account trial by trial the average response rates. In addition, the authors make the claim in several places that TVNS does not effect effort maintenance. However, on page 16 they state “Conversely, taVNS did not significantly enhance effort maintenance compared to sham stimulation ($b = 1.21$, $t = 1.715$, $p = .090$, $BF_{10} = 0.51$; Table S.2)”. Unless I am misunderstanding this would be in the range of inconclusive, with evidence neither supporting nor against the null hypothesis. In which case the authors should probably remove strong interpretations relating to which aspect of invogiration/effort that they showed effects on that are present throughout the manuscript and abstract.

3. Exhaustion – On a related note the authors state little relationship to exhaustion on a trial and invigoration (nor any effect of stimulation). However, did the authors look at whether exhaustion ratings would predict (i) the change in exhaustion from one trial to the next rather than the raw ratings (this was unclear) and (ii) whether exhaustion ratings were predictive of the next trials invigoration. It is plausible that exhaustion occurs as a sensation at the end of the trial that impacts on the next trial; rather than something that would have impacted performance on the trial that had just been completed.

4. Modeling results – To be honest the modeling aspect of the paper feels a little underdeveloped for a number of reasons (see below). It feels tagged on and I am not sure how much it adds in its current form. I also don’t think having modeling is necessary in this particular manuscript for it to be interesting.

a. All of the details are in supplementary information and they say that they use the model of Manohar et al., (2015), but this was a model of eye movements, not repeated button presses, and so much more detail in the text is needed about how the model was adapted, rather than just having equations in supplementary text (which are not described in sufficient detail to be easily understood).

b. Although model simulations are provided (figure 6), this analysis is very qualitative. Many equations could give rise to such patterns, and so it feels insufficient. Typically such models would be fit to the data and compared to alternative/simpler models where the necessity of including parameters in the model is tested against alternatives. This is necessary because it is plausible that a different model might on and off stimulation for example – which would undermine the comparisons between parameters.

c. How well does the model do at predicting behavior? – There seems to be little evidence of this provided only comparison in model parameters on and off stimulation. What would it mean if these parameters were different but the model only explained 2% of the variance in behavior?

5. The Discussion– The discussion of the results talks to a significant degree about neuromodulatory systems, dwelling on dopamine and serotonin. However, much of the discussion goes significantly beyond the data and many more obvious links to the work are not discussed.

a. The manuscript doesn’t really consider why signals to/from the body might significantly modulate

motivation – other than these signals could somehow find their way into the dopaminergic system. Yet, there is a whole literature on how motivation and interoceptive bodily signals might interact, with evidence that physical effort is modulated by the ability to accurately detect signals from the body (Herbert et al., 2007, *Psychophysiology*; Dunn et al., 2010, *Psych. Sci.*; Ainley et al., 2016, *Phil. Trans. Roy. Soc. B*).

b. I think much of the discussion around dopamine seems very speculative and given that it is not manipulated here, doesn't warrant such lengthy coverage.

c. Beyond 5b, the dopamine may not even be the most obvious candidate for the effects of vagal nerve stimulation. Inputs from the vagus nerve are very strong to the Locus coeruleus, containing a high concentration of neuroadrenaline, a neurotransmitter linked to parasympathetic function and neurons in this region linked to invigoration (Varazzani et al., 2015, *J.Neuro*). This would in fact seem a more obvious candidate for the effects than those proposed. However, linking any of this to neuromodulatory systems is all still speculative at this stage.

6. Unless I am mistaken, it is not stated how many rests were taken, nor when those rests occurred in trials as a function of difficulty or reward, only the duration of rests. Did participants take more than one rest? Did this differ between conditions? Did rests occur earlier or later in the trials? Did this change with stimulation? All of this could interestingly interact and somewhat change the interpretation of the task.

Minor

7. Usage of terminology – To be honest I found the manuscript hard to follow in places, due to the swapping and changing of terms like “invigoration”, “effort”, effort maintenance, “Maintaining higher effort”. I think the manuscript would better off simply defining terms, how it relates to analysis measures (which were also a little hard to follow), and then sticking to one set of terms throughout in the text and in the figures (which again sometimes used a different terminology). Moreover I found it confusing that “effort maintenance” which is the average number of button presses in a trial is not considered “vigor”. I think the terminologies here sometimes do not track what is used in other work in the field (for instance how Niv et al., 2007 suggest that greater movement vigor is doing an action faster repeatedly).

8. Assumed knowledge of the reader – In general the manuscript is written very much for an expert audience, and at times in an unclear manner, where terms are used having never been defined clearly. Some examples but not all assumed knowledge in the text includes:

a. Page 11 – The model comparison for mixed effects models is completely unclear. “The extended model of effort maintenance showed that it fit the data slightly, but significantly better ($p = .015$), whereas the restricted model was equivalent in fit compared to the complex model for invigoration and should be preferred ($p > .50$)” At this point in the manuscript “restricted model” etc. haven't been defined. I am still unclear exactly what this all means.

b. page 11– “Maintenance of effort was operationalized as the average frequency of button presses during a trial which is equivalent to the area under the curve” – The area under what curve? What is the reader equating it to?

9. Page 14 – “Higher difficulty reduced vigor, $b = -2.44$, $t = -3.26$, $p = .002$, but the effect of costs on invigoration was only half compared to the effect of benefits” I am not sure that this comparison of the size of effects is particularly meaningful. If difficulty levels of 30% and 90% had been used instead, the effect of effort may have been larger, vice versa if 1 cookie vs 2 cookies had been the reward levels the size of ht effects could have been flipped.

Reviewer #2 (Remarks to the Author):

In this manuscript, Neuser and colleagues examine the impact of unilateral transcutaneous auricular vagus nerve stimulation (taVNS) on motivated effortful behavior, in a task that involves motor tapping for money or food reward. The main finding is that taVNS boosts the invigoration of behavior, defined as the speed of tapping frequency rise up to the imposed target. The authors then intend to better specify this result, showing an absence of effect on effort maintenance, an effect of stimulation side (left taVNS boosting vigor for food reward only), and a change in the conversion rate from wanting (assessed through self-report) to vigor.

This study has many strengths. To my knowledge, the motivational effects of taVNS have not been investigated in humans, with a task that properly assesses the trade-off between reward and effort (although button pressing may not be as effortful as handgrip squeezing). This is important given the clinical applications of taVNS, notably for the treatment of major depression. Other positive points are the good sample size and the sham-controlled, randomized cross-over design.

Overall this is a timely, interesting and well executed study. I nevertheless have concerns about the interpretation. More precisely, I am convinced by the main result (taVNS enhances the vigor of motor tapping) but not by further specifications.

My main concern is that the effect put forward by the authors seems like a pure motor effect, and not a motivational effect. This is because

1° The speed of effort ramping up has typically been interpreted as a motor feature (independent from reward), correlated to motor symptoms in Parkinson’s disease, and shown to depend upon dopamine levels (see for instance Le Bouc and colleagues, *J Neurosci* 2016). An effect on effort maintenance (which also depends on wanting) would have been better evidence for a motivational impact of taVNS.

2° There was no effect of taVNS on wanting, a proper motivational measure, which presumably reflects the value of effort. As the authors show, taVNS seems to increase the motor output for a same level of

wanting.

3° The effect of taVNS on vigor is best captured by an additive parameter that does not interact with reward in the computational model. The interpretation given is that this parameter adjusts the reward value, since it appears in the benefit component of the expected value computation. However, it could be transposed to the cost component, in a mathematically equivalent way. An equally plausible interpretation is therefore that the parameter adjusts motor control processes.

A related issue is that the dissociation between effort invigoration and maintenance is not convincing. There is obviously a trend for an effect on effort maintenance in Figure 3, so the interaction between stimulation and measure may not be significant. Testing the effects separately is not enough because it leads to interpret an absence of effect (on effort maintenance). Similarly, the dissociation between left and right taVNS is not convincing, because it would require a significant three-way interaction (stimulation x side x reward type). These conclusions would thus need further evidence to be validated. Minor concerns:

- The articulation with pre-clinical literature is hard to follow. There seems to be a contradiction between animal losing weights following VNS and the prediction that VNS should enhance motivation for food reward. The rationale for this prediction should be better explained in the introduction.
- The articulation with clinical practice would need more elaboration. What is tested in this paper is an acute effect of taVNS, with short pulses (30 sec), while clinical protocols use long pulses (typically 30 min twice a day, 5 to 7 days per week). Moreover, the clinical effects usually appear after several weeks or months of taVNS. The difference between acute and chronic effects should be discussed, as this may be a serious limitation to the translation of the results in clinical settings.
- I like the computational model by Manohar and colleagues but it was meant to account for the speed/accuracy trade-off in saccadic movements. The noise reduction cost seems irrelevant here so I suggest using a more standard model of effort discounting, which would enable more straightforward interpretation.
- There are unnecessary details in the methods, regarding the experimental design and data analysis, that could go to supplementary information. I reckon the main text should be better focused on the main conclusions. Some figures are also unnecessarily complicated (for instance, the use of heatmaps) and could be simplified to better illustrate the results. Finally, the text would benefit from a more careful check, as there are remaining typos that sometimes confuse the explanation. For example, in Figure 5 legend, the sentence starting with "Compared to permuted data ..." does not make sense.

Reviewer #3 (Remarks to the Author):

In this study, Neuser and colleagues evaluated the effects of stimulating the auricular branch of the human vagus nerve (transcutaneous auricular vagus nerve stimulation, "taVNS") on reward-guided behaviors. The authors report significant effects of vagal stimulation on invigoration, but less so on maintenance, of reward-seeking behaviors. Whereas effects of left vagus stimulation were pronounced for food reward seeking, right vagus stimulation seemed to generalize the invigorating effects to monetary rewards. The authors concluded that taVNS enhances reward-seeking by boosting vigor, opening the possibility for taVNS use in emotional and mood disorders.

Stimulation of the vagus nerve is increasingly becoming a critical therapeutic approach to a number of disorders, in addition to its well-established use in major depression. As its mechanisms of action on the human brain remain largely unknown, the current report may provide a seminal contribution. There are however some important clarifications that will need to be provided by the authors as follows.

- The bulk of the conclusions appear to derive from the data depicted in Figure 3. While the putative effect of taVNS is apparent, it is unclear the extent to which they derive from variations in responses to sham stimulation, as opposed to taVNS per se. For example, for the Left Ear Vigor condition, there are large differences in response to sham stimulation in terms of reward type (food vs. money). For the Maintenance condition, there are large differences in responses to sham stimulation related to the side of stimulation. It is unclear why this was the case, and the manuscript does not indicate that the authors have paid attention to these variations.

- The above raises the possibility that sham stimulation is itself producing unintended effects and may therefore not be an actual "sham" condition. This may be due to the authors aiming at producing "mild pricking" in each condition to match the subjective experience of the stimulation (see page 6). In particular, it is possible that sham stimulation producing "mild pricking" on the ear lobe may have interfered with vagal function itself (e.g. by spreading current to the neighboring vagal auricular branch). This point needs to be fully clarified.

- Also related to the above is the possibility that the authors may have missed on reward seeking effects that could be obtained only upon stimulation regimes that aren't accessible to somatic perception, such as sub-threshold for "mild pricking".

- Regarding the money+food reward seeking effects associated with right vagus stimulation, was the overall main effect of stimulation significant for the right ear (Figure 3a, Right ear condition). Please state values accordingly.

Reviewer #4 (Remarks to the Author):

The manuscript by Neuser et al. is one of the many studies published in the last few years investigating the effect of tVNS on cognitive processes. The only original aspect is that, compared to other studies,

the authors stimulated both the left or the right ear. This study showed that tVNS enhances reward-seeking by boosting vigor. The results are potentially interesting, but the theoretical background is very weak and seems to be constructed post-hoc according to the results.

The authors stated “Vagal afferent activation can thereby indirectly modulate key brain circuits involved in reward (Tellez et al., 2013) and energy homeostasis (see de Lartigue, 2016) as endogenous stimulation of the gut with nutrients evokes dopamine release in the dorsal striatum tracking caloric load (de Araujo et al., 2012; Tellez et al., 2016).” This is very weak, indirect and highly speculative. Imaging studies investigating the effect of tVNS on the brain show no effect at all on the dorsal striatum (Dietrich et al., 2008; Frangos, Ellrich, & Komisaruk, 2015; Kraus et al., 2013; Yakunina et al., 2017) but tVNS activates the nucleus of the solitary tract, bilateral spinal trigeminal nucleus, dorsal raphe, locus coeruleus, and contralateral parabrachial area, amygdala, and nucleus accumbens. Accordingly, the following statement “human neuroimaging studies using fMRI have shown enhanced activity in... striatum after concurrent taVNS (Frangos, Ellrich, & Komisaruk, 2015; Kraus et al., 2007)” is not accurate. The authors assumed that the tVNS has an effect on dopamine, but there is no scientific proof of that. Instead VNS enhances the level of NA and GABA. There is plenty of evidence instead about the link iVNS augmented the release of NA in the LC and hippocampus (Dorr and Debonnel, 2006; Follesa et al., 2007; Groves et al., 2005; Hassert et al., 2004; Hulsey et al., 2017; Raedt et al., 2011; Roosevelt et al., 2006). Specifically, about tVNS salivary alpha-amylase (sAA) levels have been proposed to be an indirect marker for central NA system activation (Chatterton, Vogelsong, Lu, Ellman, & Hudgens, 1996; Warren, van den Brink, Nieuwenhuis, & Bosch, 2017). Two auricular tVNS studies have reported increased sAA levels in the active stimulation compared to sham condition (Ventura-Bort et al., 2018; Warren et al., 2019), which lends additional support to the assumption that tVNS modulates the NA system. About GABA a single-proton emission computed tomography study carried out in epileptic patients revealed that iVNS enhanced GABA receptor density in frontal and frontotemporal areas (Marrosu et al., 2003). Further, the short-interval intracortical inhibition, a transcranial magnetic stimulation paradigm indicative of GABA-A activity, was substantially enhanced in right motor cortex after active tVNS (Capone et al., 2015).

If the authors had a priori hypothesis about dopamine why they did not measure dopamine levels via blood samples? Without any dopaminergic marker to sustain their hypothesis, the theoretical background proposed by the authors is very shaky.

The authors stated: “Here, we tested whether non-invasive taVNS—applied to emulate interoceptive feedback signals emitted from the gut—would modulate instrumental behavior that is working for rewards (food or money).”

This prediction would make sense in the case of cervical tVNS (or iVNS), but not in the case of tVNS. The use of auricular and cervical tVNS might have specific effects on the neuromodulation of cognitive processes because these tVNS approaches rely on the activation of only afferent (auricular tVNS) or both afferent and efferent fibers (cervical tVNS). Given that tVNS activates only afferent fibers it is not possible for this method to emulate interoceptive feedback signals emitted from the gut because this device does not mediate any effects coming from or getting to the gut.

Response letter

We thank all reviewers for their careful review and valuable comments on our manuscript. We are positive that addressing these concerns such as providing a more detailed description of the methods and results has substantially improved the submission.

Reviewer #1:

Neuser and colleagues present a vagal nerve stimulation study in healthy people, aimed at understanding how motivation to exert effort for reward may be modulated by signal from the body. They use a modified version of a previously implemented effort allocation task, examining how people decide when to work and rest in terms of repeated button presses. They offered participants different types of rewards (food/money), at different magnitudes, with different levels of difficulty (button presses) for obtaining those rewards. They applied vagal nerve stimulation to the left ear in one group, right ear in the other and sham in both groups and examined the effect on the maintenance of effortful exertion and the initial invigoration of movements following periods of rest. Overall, they find that stimulation increases the vigor towards rewards. In addition, the authors argue that this effect may be greater for food rewards than monetary and only effects the maintenance of effort not the invigoration.

This is clearly a highly novel study. They have used a sophisticated study design, reasonable sample sizes, and I believe the primary conclusion will be of great interest to a number of different sub-fields of psychology and neuroscience. However, I do have a number of concerns about the study, which come about largely because the authors have drawn many of their inferences from what should be considered inconclusive non-significant effects, which may arise due to some methodological limitations. Many of these concerns could however be alleviated by the authors simply drawing their conclusions more from the significant effect of TVNS, which is still novel and of interest.

Major

1) Effects of TVNS on different types of reward – I can appreciate the authors desire to examine whether the effects of stimulation might enhance the wanting and motivation for primary reward, which could link to the literature on embodiment etc. However, there are several differences in this experiment between these two types of rewards that serve as a significant limitation for trying to make specific claims. For instance, participants were in a partially fasted state, with the aim of incentivising the wanting and seeking of the primary rewards, but there was no comparable approach for the monetary rewards. Thus, an equally plausible alternative interpretation is that vagal nerve stimulation increases motivation for reinforcers when they have been significantly depleted. I can completely understand the rationale for taking this approach and appreciate you cannot similarly make your participants experience poverty. However, I think this limitation impairs the ability to make strong claims about the specificity of the vagal nerve effects on reward types, but is strongly interpreted

throughout the manuscript. But I also think that for such a novel study the strong claims on specificity are also not necessary for this study to still be of interest.

We thank the reviewer for the detailed consideration of our manuscript and the thoughtful comments. Monetary rewards are often used in human experimental work precisely because they are not subject to an equivalent of fasting/satiety. Sensitivity to depletion is less often considered, however, most likely for reasons that the reviewer has already pointed out. In our experiment, we used food rewards as the second reward type for several reasons. First, previous studies on motivational effects in rodents used primarily food rewards so it was not clear whether the effects would generalize to money. Second, we expected vagal afferents to forward interoceptive signals related to energy metabolism, so we were looking for an opportunity to contrast taVNS-induced effects on a reinforcer that is subject to satiety (food) vs. one that is not (money). Relatedly, we invited participants in a hungry state to ensure a high incentive value of prospective food rewards for every participant. Nevertheless, we agree with the reviewer that we cannot resolve whether taVNS acts more broadly to depletion and we have revised the discussion and limitations section to better reflect that possibility.

“Fourth, although we selected food versus monetary rewards to assess generalization to secondary reinforcers, it is conceivable that other characteristics of the two rewards such as depletion status could affect taVNS-induced effects on invigoration. To better understand the precise behavioral mechanism, it is advisable to extend the set of reinforcers and conditions such as the status of resource depletion.” (p.25)

2) Specificity of effects in regards to aspects of “effort” or invigoration. A similar issue to point 1 arises when the claims are made in the paper regarding the effects of TVNS on the two measures of effort. Firstly, it is slightly odd, in a way, that they are treated as separate independent measures, because one measure effects the other within a trial. However, the two measures are not put together in the same analysis i.e. testing whether “invigoration” is modulated by TVNS taking into account trial by trial the average response rates. In addition, the authors make the claim in several places that TVNS does not effect effort maintenance. However, on page 16 they state “Conversely, taVNS did not significantly enhance effort maintenance compared to sham stimulation ($b = 1.21$, $t = 1.715$, $p = .090$, $BF_{10} = 0.51$; Table S.2)”. Unless I am misunderstanding this would be in the range of inconclusive, with evidence neither supporting nor against the null hypothesis. In which case the authors should probably remove strong interpretations relating to which aspect of invigoration/effort that they showed effects on that are present throughout the manuscript and abstract.

We thank the reviewer for pointing out that presenting the two outcome measures as if they are entirely independent was an oversight. Of course, we had evaluated the correlation of the two measures, which was perhaps surprisingly moderate at the trial level ($r = .286$, 95% CI [.254, .321]). Therefore, we decided that presenting the two mixed-effects models was a straightforward

way to estimate taVNS-induced effect on complementary facets of motivation. Nevertheless, we agree with the reviewer that the stimulation effects on invigoration should be evaluated in a model where we include effort maintenance in the trial-based analysis. As requested, we ran the proposed extension of the model and still observed a taVNS-induced effect on invigoration after inclusion of effort maintenance as additional predictor ($b_{adj} = 1.99, p = .025$). Notwithstanding, the taVNS effect was attenuated compared to the model not including effort maintenance ($b_{unadj} = 2.93, p = .004$). We think that this analysis supports our main conclusion that taVNS facilitates invigoration while it also supports the reviewer's idea that there is shared variance with effort maintenance adding to the observed effect of taVNS. Thus, we revised the manuscript to better reflect the inherent association of invigoration and effort maintenance by incorporating statements in the methods and results sections.

3) Exhaustion – On a related note the authors state little relationship to exhaustion on a trial and invigoration (nor any effect of stimulation). However, did the authors look at whether exhaustion ratings would predict (i) the change in exhaustion from one trial to the next rather than the raw ratings (this was unclear) and (ii) whether exhaustion ratings were predictive of the next trials invigoration. It is plausible that exhaustion occurs as a sensation at the end of the trial that impacts on the next trial; rather than something that would have impacted performance on the trial that had just been completed.

We thank the reviewer for these excellent suggestions on the relation of 'objective' (i.e., effort maintenance) and 'subjective' measures of effort exertion.

First, we would like to emphasize that the wording of the VAS item presented after each trial was "Wie stark haben Sie sich in diesem Durchgang verausgabt?" ["How much did you exert your resources in the past trial?"]. In the manuscript, we referred to this item as an "exhaustion" rating, but after discussions with several native English speakers, who are also fluent in German, it seems more appropriate to call it "exertion" ratings instead and we have now changed the label accordingly throughout the manuscript.

Second, there was little evidence that the task produced cumulative exhaustion/exertion as we did not observe significant decreases, but rather minor improvements in effort maintenance over trials ($b = 0.047, p = 0.010$) and no change in exertion ratings over trials ($b = 0.033, p = .174$; estimation of random slope coefficients for a trial regressor in a full mixed-effects model).

Third, per request of the reviewer, we tested whether exertion ratings on the previous trial predict invigoration on the next trial. However, exertion ratings in the preceding trial were not associated with invigoration in the following trial ($b = 0.017, p = .414$). Notably, exertion ratings in the preceding trial were negatively associated with effort maintenance in the next trial (accounting for wanting and exertion ratings on the same trial, $b = -0.062, p < .001$). This suggests that exertion on the previous trial may play a limited (i.e., the b is relatively small), yet significant role (i.e., the standard error is relatively small as well) in guiding effort maintenance. However, since invigoration was not

associated with rated exertion on the same trial or the previous trial, this supports our conclusion that it is less associated with the cost component of effort exertion.

4) Modeling results – To be honest the modeling aspect of the paper feels a little underdeveloped for a number of reasons (see below). It feels tagged on and I am not sure how much it adds in its current form. I also don't think having modeling is necessary in this particular manuscript for it to be interesting.

a) All of the details are in supplementary information and they say that they use the model of Manohar et al., (2015), but this was a model of eye movements, not repeated button presses, and so much more detail in the text is needed about how the model was adapted, rather than just having equations in supplementary text (which are not described in sufficient detail to be easily understood).

b) Although model simulations are provided (figure 6), this analysis is very qualitative. Many equations could give rise to such patterns, and so it feels insufficient. Typically such models would be fit to the data and compared to alternative/simpler models where the necessity of including parameters in the model is tested against alternatives. This is necessary because it is plausible that a different model might on and off stimulation for example – which would undermine the comparisons between parameters.

c) How well does the model do at predicting behavior? – There seems to be little evidence of this provided only comparison in model parameters on and off stimulation. What would it mean if these parameters were different but the model only explained 2% of the variance in behavior?

We thank the reviewer for arguing convincingly that the Manohar et al. model does not add much in its current form. Upon reflection, we agree that some of the issues cannot be addressed without a fundamental revision and extension of the model and that delivering on this end would require much more extensive computational analyses, which, in the end, would go beyond the scope of our study and not add essential new insights to our manuscript. In light of our emphasis on the novel experimental results and the previously established cost-evidence accumulation model, we removed the part regarding the Manohar et al. “cost of control” model and instead focus on cost-evidence accumulation in the main body of the manuscript. Of note, we also extended the estimation of the cost-evidence accumulation model and used a hierarchical Bayesian estimation to directly estimate the support for the alternative versus the null hypothesis. We believe that strengthening the results section about the previously established model also strengthens the paper overall.

5) The Discussion– The discussion of the results talks to a significant degree about neuromodulatory systems, dwelling on dopamine and serotonin. However, much of the discussion goes significantly beyond the data and many more obvious links to the work are not discussed.

a) The manuscript doesn't really consider why signals to/from the body might significantly modulate motivation – other than these signals could somehow find their way into the dopaminergic system. Yet, there is a whole literature on how motivation and interoceptive bodily signals might interact, with evidence that physical effort is modulated by the ability to accurately detect signals from the body (Herbert et al., 2007, *Psychophysiology*; Dunn et al., 2010, *Psych. Sci.*; Ainley et al., 2016, *Phil. Trans. Roy. Soc. B*).

We thank the reviewer for the suggestion to broaden the perspective on the effects that we observed in our experiment. We agree that interoceptive signals play an important role in modulating physical effort and that these findings may provide an important perspective on our findings. Our primary hypothesis was largely informed by seminal research on vagus nerve stimulation in animals (Han et al., 2018). Therefore, our goal was to test such an association in humans although we agree that further data is needed to establish the exact neural mechanism. Moreover, we focused more on the dopamine system in the discussion because taVNS primarily affected invigoration, not effort maintenance. Nevertheless, we have now included a statement in the discussion to refer to this possible mechanism because we agree that there are multiple viable mechanisms that will need to be resolved in the future:

“Furthermore, taVNS-induced effects on invigoration could also be partly driven by an alteration of “bodily precision” (Ainley, Apps, Fotopoulou, & Tsakiris, 2016), that is by modulating the sensitivity to interoceptive signals that guide motivation (Herbert, Ulbrich, & Schandry, 2007) and decision making (Dunn et al., 2010).” (S. 23f)

Crucially, we made more extensive revisions throughout the manuscript to improve the balance of our theoretical considerations.

b) I think much of the discussion around dopamine seems very speculative and given that it is not manipulated here, doesn't warrant such lengthy coverage.

We agree that a direct demonstration of taVNS-induced dopaminergic changes are necessary to provide conclusive evidence on the putative role of changes in dopamine in humans. Although we are positive that our findings provide an important premise for follow-up research on the exact changes in neurotransmission, we revised the discussion to focus to a greater extent on alternative pathways and hope that this will facilitate future work on the neural mechanism of taVNS. Beyond 5b, we will further elaborate on our rationale in the reply to 5c.

c) Beyond 5b, the dopamine may not even be the most obvious candidate for the effects of vagal nerve stimulation. Inputs from the vagus nerve are very strong to the Locus coeruleus, containing a high concentration of neuroadrenaline, a neurotransmitter linked to parasympathetic function and neurons in this region linked to invigoration (Varazzani et al., 2015, *J.Neuro*). This would in fact seem a more obvious candidate for the effects than those

proposed. However, linking any of this to neuromodulatory systems is all still speculative at this stage.

We agree that changes in dopamine are only one plausible candidate mechanism and that changes in other monoaminergic neurotransmitters such as noradrenaline (NA) may also lead to alterations in invigoration or motivation. Notwithstanding, it is also not clear whether reported changes in NA after VNS are partly due to changes in dopamine transmission because the synthesis of NA is dependent on dopamine levels. In the revised version, we have taken great care to give a more balanced account of current findings as we wholeheartedly agree with the reviewer that better data on neurotransmission will be necessary to settle these speculations. Our focus on dopamine was therefore also deliberate because most of the human VNS literature does not consider dopamine as a potential signaling pathway despite the clear evidence provided by animal research. All in all, we rephrased the discussion to encourage more work on this very important issue and think that it is most likely that taVNS has diverse effects on neurotransmission.

6) Unless I am mistaken, it is not stated how many rests were taken, nor when those rests occurred in trials as a function of difficulty or reward, only the duration of rests. Did participants take more than one rest? Did this differ between conditions? Did rests occur earlier or later in the trials? Did this change with stimulation? All of this could interestingly interact and somewhat change the interpretation of the task.

We agree with the reviewer that rest phases are an interesting outcome measure by itself although we did not further interrogate the data after we observed little indication of a taVNS effect on the cost component of motivation. During sham stimulation, participants had an average of 4.47 ($SD = 2.06$) rest phases per trial which includes an initial rest phase at the onset of the trial. During taVNS, participants had an average of 4.45 ($SD = 2.17$) rest phases per trial. Rest phases were more likely to occur during later stages of a trial, but, again, there was little indication of a stimulation effect (see Figure S.3).

Figure S.3: Similar temporal profiles of effort over time for both stimulation conditions. Although effort was slightly higher overall for taVNS, there was no taVNS-induced change in the timing of work and rest phases apparent at a group level. To depict the heat map, data was binned by second and averaged for each condition.

Minor

7) Usage of terminology – To be honest I found the manuscript hard to follow in places, due to the swapping and changing of terms like “invigoration”, “effort”, effort maintenance, “Maintaining higher effort”. I think the manuscript would better off simply defining terms, how it relates to analysis measures (which were also a little hard to follow), and then sticking to one set of terms throughout in the text and in the figures (which again sometimes used a different terminology). Moreover, I found it confusing that “effort maintenance” which is the average number of button presses in a trial is not considered “vigor”. I think the terminologies here sometimes do not track what is used in other work in the field (for instance how Niv et al, 2007 suggest that greater movement vigor is doing an action faster repeatedly).

We appreciate the reviewer’s emphasis on the consistent use of terms to improve the readability of our manuscript. Hence, we revised the manuscript once again in an attempt to improve the consistent use of the terms. We refer to the two outcomes as invigoration and effort maintenance. We would like to point out that vigor is necessary for both invigoration and maintenance due to the task requirement to quickly and repeatedly perform button presses. However, the emphasis on maintenance is intended because it highlights an additional component of vigor related to enduring effort. Quickly performing button presses for a limited time only would be considered vigorous and lead to a steep invigoration slope. Yet, due to the short time, this would only have a minor effect on the earned points as wins are heavily dependent on maintaining vigor over time. In other words, to do the task well, you need to

work vigorously. A fast invigoration will get you above the line more quickly thereby reducing opportunity cost. However, if you cannot maintain vigorous responding over longer periods, it will not accumulate a lot of points.

8) Assumed knowledge of the reader –In general the manuscript is written very much for an expert audience, and at times in an unclear manner, where terms are used having never been defined clearly. Some examples but not all assumed knowledge in the text includes:

a. Page 11 – The model comparison for mixed effects models is completely unclear. “The extended model of effort maintenance showed that it fit the data slightly, but significantly better ($p = .015$), whereas the restricted model was equivalent in fit compared to the complex model for invigoration and should be preferred ($p > .50$)” At this point in the manuscript “restricted model” etc. haven’t been defined. I am still unclear exactly what this all means.

We are sorry that we did not elaborate on some of the terms that we used, particularly in the data analysis section. As part of the thorough revision of the manuscript, we asked non-expert test readers to identify sections that might be difficult to comprehend and added clarifications throughout the manuscript to avoid ambiguity or jargon. Specifically, regarding the model comparisons section, we rephrased the section:

“Using nested model comparisons (χ^2), we also assessed whether sex and BMI should be included as nuisance regressors along stimulation side and order. Put simply, any additional regressor will improve the fit of a model, but this improved fit comes at the cost of increased complexity. To guard against overfitting, we used model comparisons to evaluate whether the additional complexity of an extended model is justified. For effort maintenance, the extended model (containing four participant level-predictors) fit the data slightly, but significantly better ($p = .015$). In contrast, for invigoration, the restricted model (containing only two participant-level predictors) was equivalent in fit compared to the extended model and should be preferred because it is parsimonious ($p > .50$). Since the extended models (including sex and BMI as additional covariates) led to slightly lower p-values of the stimulation effect without changing the conclusions, we report the restricted model for invigoration and maintenance to aid direct comparisons.” (p.11)

b. page 11– “Maintenance of effort was operationalized as the average frequency of button presses during a trial which is equivalent to the area under the curve” – The area under what curve? What is the reader equating it to?

We thank the reviewer for pointing out that this is not sufficiently clear. Since we sampled the estimated relative frequency at 100 ms for the analysis, the area under the curve would correspond to the area covered by a time series of the relative frequency (where time is on the x-axis and relative frequency is on the y-axis). Mathematically, integrating over this discrete time series is equivalent to a sum over each time point. Dividing this sum by the number of time bins therefore yields the average relative frequency (i.e., a normalized

measure of effort). Adding the statement on the area under the curve was an attempt to make it more intuitive for researchers, who are using the area under the curve to measure cumulative aspects such as drug levels. However, after some of our test readers had questions about this comparison as well, we revised the statement to improve comprehensibility:

“Maintenance of effort was operationalized as the average frequency of button presses during a trial capturing how much effort participants produce over time.” (p.10)

9) Page 14 – “Higher difficulty reduced vigor, $b = -2.44$, $t = -3.26$, $p = .002$, but the effect of costs on invigoration was only half compared to the effect of benefits” I am not sure that this comparison of the size of effects is particularly meaningful. If difficulty levels of 30% and 90% had been used instead, the effect of effort may have been larger, vice versa if 1 cookie vs 2 cookies had been the reward levels the size of the effects could have been flipped.

We agree with the reviewer that the effect size is arguably dependent on the given scale of the factors. Still, despite this limitation, we think that providing a more intuitive description of the differences within the task setup as it was implemented is still useful. Our primary purpose was to contrast the differential contributions of difficulty to the two primary outcomes invigoration and effort maintenance. In this case, the somewhat arbitrary setting of the factor levels is less of a concern.

Moreover, at a more conceptual level, the difficulty levels were carefully selected after using a continuous set of difficulty levels in a pilot. We set the low difficulty condition to 75% so that participants would be able to maintain effort for about 20 s out of 30 s. For the high difficulty condition, we used 85% so that participants would be able to maintain effort for about 10 s out of 30 s, leading to more rest phases. In that way, the difficulty setting was not entirely arbitrary and reflected a useful range of behavior within the task setup.

Reviewer #2

In this manuscript, Neuser and colleagues examine the impact of unilateral transcutaneous auricular vagus nerve stimulation (taVNS) on motivated effortful behavior, in a task that involves motor tapping for money or food reward. The main finding is that taVNS boosts the invigoration of behavior, defined as the speed of tapping frequency rise up to the imposed target. The authors then intend to better specify this result, showing an absence of effect on effort maintenance, an effect of stimulation side (left taVNS boosting vigor for food reward only), and a change in the conversion rate from wanting (assessed through self-report) to vigor.

This study has many strengths. To my knowledge, the motivational effects of taVNS have not been investigated in humans, with a task that properly assesses the trade-off between reward and effort (although button pressing may not be as effortful as handgrip squeezing). This is important given the clinical applications of taVNS, notably for the treatment of major depression. Other positive points are the good sample size and the sham-controlled, randomized cross-over design.

Overall this is a timely, interesting and well executed study. I nevertheless have concerns about the interpretation. More precisely, I am convinced by the main result (taVNS enhances the vigor of motor tapping) but not by further specifications.

My main concern is that the effect put forward by the authors seems like a pure motor effect, and not a motivational effect. This is because

1) The speed of effort ramping up has typically been interpreted as a motor feature (independent from reward), correlated to motor symptoms in Parkinson's disease, and shown to depend upon dopamine levels (see for instance Le Bouc and colleagues, *J Neurosci* 2016). An effect on effort maintenance (which also depends on wanting) would have been better evidence for a motivational impact of taVNS.

We thank the reviewer for his or her careful evaluation of our manuscript and the suggestions. We agree that the speed of invigoration is correlated with symptoms in Parkinson's disease and dependent on dopamine levels (Mazzoni, Hristova, & Krakauer, 2007; Panigrahi et al., 2015). Nevertheless, there is good evidence that the invigoration is, in fact, dependent on reward magnitude (Bühler et al., 2010; Kroemer et al., 2014; McGinty, Lardeux, Taha, Kim, & Nicola, 2013). We also demonstrate this association in our task because we observed that invigoration was greater for higher reward magnitudes: "participants were quicker to invigorate behavior when more reward was at stake, $b = 5.79$, $t = 4.69$, $p < .001$ (Fig. 2a)".

Furthermore, invigoration was associated with wanting ratings, but not ratings of exertion during the same trial (Fig. 2e) or exertion during the previous trial. In our opinion, this indicates that faster approach of rewards is a motivational characteristic, not a simple motor feature that is independent of motivational aspects. Relatedly, it has been argued that motivational aspects of motor control crucially contribute to the motor symptoms in Parkinson's disease

(Mazzoni et al., 2007) and that this is modulated by alterations in dopaminergic transmission (Chong et al., 2015; Le Heron et al., 2018).

2) There was no effect of taVNS on wanting, a proper motivational measure, which presumably reflects the value of effort. As the authors show, taVNS seems to increase the motor output for a same level of wanting.

We thank the reviewer for raising this important question. We agree that changes in wanting would point to changes in a common motivational measure. Nevertheless, as we have discussed before, invigoration is an important motivational measure in its own right. Illustratively, we may consider a group of students who would all want to have a perfect grade. Despite the absence of differences in subjective wanting, they may differ in their vigor to pursue goal-directed behavior such as going quickly to the library to get a book on the topic that will be covered in the exam. Relatedly, we would like to refer to a recent perspective, where de Araujo, Schatzker, and Small (2019) have argued that reward signals originating from the gut may act subconsciously, leading to changes in behavior that are not instantly reflected in subjective ratings. Consequently, we consider the behavioral outcome invigoration as an important indication of a change in a motivational outcome that is complementary to wanting.

3) The effect of taVNS on vigor is best captured by an additive parameter that does not interact with reward in the computational model. The interpretation given is that this parameter adjusts the reward value, since it appears in the benefit component of the expected value computation. However, it could be transposed to the cost component, in a mathematically equivalent way. An equally plausible interpretation is therefore that the parameter adjusts motor control processes.

We agree with the reviewer that there are mathematically equivalent ways to set up the model so that it adjusts primarily motor control processes. To clarify, we would like to point out that increases in invigoration could be driven by changes in benefits or costs or even both and the model alone cannot resolve this question. Due to several issues with our previous approach to simulate changes, we decided to remove this section and to replace it with an extended modeling section on the cost-evidence accumulation model.

4) A related issue is that the dissociation between effort invigoration and maintenance is not convincing. There is obviously a trend for an effect on effort maintenance in Figure 3, so the interaction between stimulation and measure may not be significant. Testing the effects separately is not enough because it leads to interpret an absence of effect (on effort maintenance). Similarly, the dissociation between left and right taVNS is not convincing, because it would require a significant three-way interaction (stimulation x side x reward type). These conclusions would thus need further evidence to be validated.

We thank the reviewer for these constructive suggestions. We agree that there is a trend for an effect on effort maintenance. Nevertheless, the effect of taVNS on invigoration is still significant even if we account for effort maintenance on the same trial:

See our response to Reviewer #1 (Comment 2):

“We thank the reviewer for pointing out that presenting the two outcome measures as if they are entirely independent was an oversight. Of course, we had evaluated the correlation of the two measures, which was perhaps surprisingly moderate at the trial level ($r = .286$, 95% CI [.254, .321]). Therefore, we decided that presenting the two mixed-effects models was a straightforward way to estimate taVNS-induced effect on complementary facets of motivation. Nevertheless, we agree with the reviewer that the stimulation effects on invigoration should be evaluated in a model where we include effort maintenance in the trial-based analysis. As requested, we ran the proposed extension of the model and still observed a taVNS-induced effect on invigoration after inclusion of effort maintenance as additional predictor ($b_{adj} = 1.99$, $p = .025$). Notwithstanding, the taVNS effect was attenuated compared to the model not including effort maintenance ($b_{unadj} = 2.93$, $p = .004$). We think that this analysis supports our main conclusion that taVNS facilitates invigoration while it also supports the reviewer’s idea that there is shared variance with effort maintenance adding to the observed effect of taVNS. Thus, we revised the manuscript to better reflect the inherent association of invigoration and effort maintenance by incorporating statements in the methods and results sections.”

Furthermore, we agree with the reviewer that a significant three-way interaction is necessary to conclude that there is a difference between the sides; this interaction is indeed significant:

“Although the side of the stimulation did not affect the main effect of taVNS ($p = .947$), taVNS on the left side led to a significantly stronger interaction effect (cross-level interaction of Stimulation Side on Stimulation \times Reward Type, $b = -2.82$, $t = -2.122$, $p = .037$). The corresponding Bayes factor did not reach a moderate evidence level, $BF_{10} = 2.40$. Nevertheless, restricting the analysis of the Stimulation \times Reward Type effect to the left side of taVNS provided strong evidence for a food-specific effect, $t = 3.172$, $p = .003$, $BF_{10} = 11.80$. In contrast, stimulation on the right side did not lead to a Stimulation \times Reward Type effect, $t = -.118$, $p = .91$, $BF_{10} = 0.17$ and provided moderate evidence against an interaction.”

Minor concerns:

5) The articulation with pre-clinical literature is hard to follow. There seems to be a contradiction between animal losing weights following VNS and the prediction that VNS should enhance motivation for food reward. The rationale for this prediction should be better explained in the introduction.

We agree with the reviewer that the effects of VNS on food intake and motivation might seem contradicting at first glance. It was precisely this contradiction that also sparked our interest initially. To better understand these effects, it is crucial to distinguish acute effects on the motivational system that effectively reinforce behavior (likely driven by dopamine (Han et al., 2018)) and chronic effects of the stimulation resembling a satiety signal (Yao et al., 2018). Notably, we have recently shown that acute taVNS also reduces gastric myoelectric frequency of the stomach (Teckentrup et al., 2020). Such a mechanism may link acute and chronic effects as a slowdown of the digestive tract may momentarily facilitate motivational approach while reducing food intake in the long run. Thus, we added to following section to the introduction:

“To control food intake, vagal afferents primarily provide negative feedback signals (Yao et al., 2018), routed via the nucleus tractus solitarii (NTS). These vagal afferent projections are sufficient as decerebrated rats still terminate meal intake (Guillaume de Lartigue, 2016). In line with this idea, chronic VNS has been consistently shown to reduce body weight in animals and humans. Preclinical studies indicate that this is primarily due to reduced food intake (Val-Laillet, Biraben, Randuineau, & Malbert, 2010; Yao et al., 2018). Likewise, two recent studies have shown that acute taVNS reduces gastric myoelectric frequency of the stomach (Hong et al., 2019; Teckentrup et al., 2020). At the same time, acute VNS has reinforcing properties leading to sustained self-stimulation and conditioning preferences for flavors or places via a dopaminergic mechanism (Han et al., 2018). Furthermore, activation of vagal afferents regulates learning and memory in rats and humans suggesting a role in reward seeking (Kühnel et al., 2019; Suarez et al., 2018). Therefore, chronic reductions in food intake could be linked to acute increases in motivational drive by a combination of afferent and efferent effects.” (p. 3f)

6) The articulation with clinical practice would need more elaboration. What is tested in this paper is an acute effect of taVNS, with short pulses (30 sec), while clinical protocols use long pulses (typically 30 min twice a day, 5 to 7 days per week). Moreover, the clinical effects usually appear after several weeks or months of taVNS. The difference between acute and chronic effects should be discussed, as this may be a serious limitation to the translation of the results in clinical settings.

We thank the reviewer for raising the issue of a potential translational gap that needs to be elaborated. To date, most studies have focused on chronic VNS protocols for clinical applications. Nevertheless, we are using the same stimulation protocol preset in the NEMOS device in terms of pulse length, duty cycle etc. that was used in clinical studies before. Accordingly, the device should be used daily for up to 4 h. Therefore, our taVNS “dose” resembles a typical day of use in the field, but arguably not its longer consecutive use. We agree with the reviewer that this difference should be discussed, but we do see this gap not only as a limitation: Ideally, one would be able to predict the

chronic effects of a motivational treatment based on its acute effects after one day of use. First, instant improvements of motivational symptoms are currently scarce and may provide a huge potential for improvement in clinical settings by reducing the lag to a potential treatment response. Second, the identification of an acute mechanism that is linked to chronic improvement could help to rapidly identify responders. Still, we are at a very early stage to evaluate the translational potential of our findings, and we agree with the reviewer that this should be pointed out very clearly:

“Notwithstanding, it is unknown to date whether acute improvements can predict sustained therapeutic effects of potential taVNS-based treatments which remains as a vital question for the translation to clinical settings.” (p. 26)

7) I like the computational model by Manohar and colleagues, but it was meant to account for the speed/accuracy trade-off in saccadic movements. The noise reduction cost seems irrelevant here so I suggest using a more standard model of effort discounting, which would enable more straightforward interpretation.

We agree with the reviewer’s concern and have decided to remove the model from the revised version of the manuscript. Instead, we focused on the cost-evidence accumulation model that was previously validated for a comparable version of the task because most effort discounting models are designed for binary choices over a larger range of difficulty and reward magnitude options (see also our response to major comment 4 of reviewer #1).

8) There are unnecessary details in the methods, regarding the experimental design and data analysis, that could go to supplementary information. I reckon the main text should be better focused on the main conclusions. Some figures are also unnecessarily complicated (for instance, the use of heatmaps) and could be simplified to better illustrate the results. Finally, the text would benefit from a more careful check, as there are remaining typos that sometimes confuse the explanation. For example, in Figure 5 legend, the sentence starting with “Compared to permuted data ...” does not make sense.

We thank the reviewer for his or her suggestions on how to improve the focus of the manuscript. We have streamlined the methods and moved some unnecessary details to the supporting information. The figure containing heat maps was also replaced. Moreover, we carefully revised the spelling throughout the manuscript as we learned once more that the best way to spot typos is to click “submit”.

Reviewer #3

In this study, Neuser and colleagues evaluated the effects of stimulating the auricular branch of the human vagus nerve (transcutaneous auricular vagus nerve stimulation, "taVNS") on reward-guided behaviors. The authors report significant effects of vagal stimulation on invigoration, but less so on maintenance, of reward-seeking behaviors. Whereas effects of left vagus stimulation were pronounced for food reward seeking, right vagus stimulation seemed to generalize the invigorating effects to monetary rewards. The authors concluded that taVNS enhances reward-seeking by boosting vigor, opening the possibility for taVNS use in emotional and mood disorders.

Stimulation of the vagus nerve is increasingly becoming a critical therapeutic approach to a number of disorders, in addition to its well-established use in major depression. As its mechanisms of action on the human brain remain largely unknown, the current report may provide a seminal contribution. There are however some important clarifications that will need to be provided by the authors as follows.

1) The bulk of the conclusions appear to derive from the data depicted in Figure 3. While the putative effect of taVNS is apparent, it is unclear the extent to which they derive from variations in responses to sham stimulation, as opposed to taVNS per se. For example, for the Left Ear Vigor condition, there are large differences in response to sham stimulation in terms of reward type (food vs. money). For the Maintenance condition, there are large differences in responses to sham stimulation related to the side of stimulation. It is unclear why this was the case, and the manuscript does not indicate that the authors have paid attention to these variations.

We thank the reviewer for his or her evaluation of the manuscript and the additional questions regarding the sham condition. First, we would like to assure the reviewer that we paid attention to these variations by contrasting taVNS sessions to sham sessions in a full mixed-effects model that allows for the estimation of random slopes (Chen, Saad, Britton, Pine, & Cox, 2013; Raudenbush & Bryk, 2002). This model does not require that the sham treatment does not cause effects by itself. Instead, we quantify whether there is systematic evidence for a stimulation effect relative to the individual performance during sham (while using group-level information to improve the estimation of the statistical model). Thus, analogous to using a placebo condition, we can basically assess what differentiates taVNS from sham without assuming that sham could not alter behavior in one way or the other, for example, by producing a mildly pricking sensation.

Regarding the first set of requested analyses, we ran an additional repeated measures ANOVA based on the individual averages of the invigoration slope separated by reward type (Money vs. Food) and reward magnitude. Whereas reward magnitude had a significant effect on invigoration during sham as well, $F(1,80)=19.977$, $p < .001$, there was no significant difference due to reward type, $F(1,80)=3.335$, $p = .072$. Thus, we agree with the reviewer that there appears to be a difference in performance when monetary vs. food rewards are at stake

during sham. However, this difference was not significant and small compared to the effect of reward magnitude.

Regarding the second set of requested analyses, we ran an additional repeated measures ANOVA based on the individual averages of the effort maintenance separated by reward type (Money vs. Food) and reward magnitude. Furthermore, we included the side of the stimulation as a between-subject factor. Again, reward magnitude had a significant effect on effort maintenance during sham, $F(1,80)=46.112$, $p < .001$, and there was no significant difference due to reward type, $F(1,80)=0.378$, $p = .540$. More importantly, there was also no significant effect of the side of the stimulation on effort maintenance (between-subject effect of stimulation side: $F(1,79) = 3.138$, $p = .080$). Likewise, we agree with the reviewer that there appears to be a difference in performance for the group of participants who received stimulation on the right side. However, this difference was not significant despite a clearly visible difference in the depicted averages. The reason for the absence of a significant effect is the considerable inter-individual variability of effort maintenance because this is not a within-subject comparison. For example, for the low reward monetary condition, $M(\text{left}) = 66.54$, $SD = 31.06$ vs. $M(\text{right}) = 56.03$, $SD = 20.98$ but this difference is small compared to the variability of individual differences, $M(\text{left+right}) = 61.35$, $SD = 18.93$.

Based on these results, we concluded that there is not enough evidence to support an effect of sham stimulation that could not be explained by sampling variance as well. Nevertheless, we agree that it is possible that sham stimulation alone could elicit changes in behavior or brain response, even if the effect size is small. Therefore, we believe that it is preferable to demonstrate the stimulation effects using a randomized cross-over design that provides individual estimates controlled for potential effects of sham stimulation, as we did, instead of between-group designs.

2) The above raises the possibility that sham stimulation is itself producing unintended effects and may therefore not be an actual "sham" condition. This may be due to the authors aiming at producing "mild pricking" in each condition to match the subjective experience of the stimulation (see page 6). In particular, it is possible that sham stimulation producing "mild pricking" on the ear lobe may have interfered with vagal function itself (e.g. by spreading current to the neighboring vagal auricular branch). This point needs to be fully clarified.

We appreciate the reviewer's concern about potential side effects elicited by our sham procedure. We used an approved medical stimulation device with the instruction to set the amplitude so that it produces mild pricking. Therefore, we matched the stimulation amplitude to the subjective sensation of "mild pricking" to improve blinding of the participants with regard to the "true" taVNS condition. Our stimulation procedure including the sham condition was based on earlier work by Frangos, Ellrich, and Komisaruk (2015). As noted before, we agree that it is conceivable that sham stimulation may produce behavioral effects. However, we reason that there is good evidence from previous studies that stimulation of vagal afferents produces effects that are more than just the absence of interference with vagal function. For example, in

a seminal study, (Fallgatter et al., 2003) reported vagus sensory evoked potential from the brainstem after stimulating the “inner side of the tragus”, but not after stimulating the earlobe. Other neuroimaging studies have shown incremental effects on brain responses of vagal afferent stimulation compared to sham stimulation in humans (Frangos et al., 2015; Frangos & Komisaruk, 2017; Sclocco et al., 2019; Yakunina, Kim, & Nam, 2017). Moreover, animal studies have shown that transcutaneous stimulation of vagal afferents elicits changes in the brain that are in line with an activation of vagal afferent projections. Notably, we have recently shown that an analogous stimulation protocol elicits changes in gastric myoelectric frequency, a known efferent effect of vagal regulation (Teckentrup et al., 2020) and this effect was primarily driven by changes during taVNS, not by changes during sham, relative to a baseline without stimulation.

3) Also related to the above is the possibility that the authors may have missed on reward seeking effects that could be obtained only upon stimulation regimes that aren't accessible to somatic perception, such as sub-threshold for "mild pricking".

We thank the reviewer for pointing out this possibility. We agree that it is possible that subthreshold stimulation has qualitatively or quantitatively discernible effects on motivation. Notably, the sensation of “mild pricking” is described in the user’s manual of the stimulation device as the correct individual stimulation strength for therapeutic use. Since one of our goals was to shed more light on potentially invigorating or motivating effects of taVNS as it is used in therapeutic settings today, we decided to follow a previously established and commonly used stimulation protocol. We added a comment to indicate that this is an open question at the moment:

“Fifth, we used a stimulation strength that produced a somatic sensation according to the instructions of the therapeutic use of the taVNS device. It is possible that different stimulation protocols could lead to different effects on motivation.” (p. 25)

4) Regarding the money+food reward seeking effects associated with right vagus stimulation, was the overall main effect of stimulation significant for the right ear (Figure 3a, Right ear condition). Please state values accordingly.

We did not look separately at the main effects of taVNS by side of the stimulation because there was no significant interaction in the model, but we are happy to provide the values in addition. As per the request of the reviewer, we reanalyzed the main effect residuals (corrected for order) by stimulation side. For the left ear, we observed a significant main effect ($b = 2.86$, 95% CI [.10, 5.48], $t = 2.193$, $p = .034$). For the right ear, we observed a marginally significant main effect ($b = 3.00$, 95% CI [.03, 5.83], $t = 2.016$, $p = .051$) that was numerically almost the same as during stimulation of the left ear and the bias-corrected and accelerated bootstrap interval did not include 0. Thus, the

absence of a significant effect for the right ear alone despite a higher b is primarily due to a slightly larger inter-individual variability.

Reviewer #4

The manuscript by Neuser et al. is one of the many studies published in the last few years investigating the effect of tVNS on cognitive processes. The only original aspect is that, compared to other studies, the authors stimulated both the left or the right ear. This study showed that tVNS enhances reward-seeking by boosting vigor. The results are potentially interesting, but the theoretical background is very weak and seems to be constructed post-hoc according to the results.

We thank the reviewer for his or her evaluation of our manuscript. We agree that there has been a surge of research on the effects of tVNS in recent years. However, to the best of our knowledge, this is the first experimental study on acute motivational effects elicited by taVNS in humans using a comparatively large randomized cross-over design and two sides of stimulation. We also agree with the concern raised by the reviewer throughout the review that potential motivational effects (allegedly modulated by dopamine) have raised little attention in the human literature, despite the mounting evidence from preclinical research. We feel that this aspect increases the relevance and novelty of our work. Crucially, we strongly dispute the suspicion that the theoretical background was “constructed post-hoc according to the results”, which we would consider as scientifically deceptive and insincere. To demonstrate our *a priori* train of thought, we have shared the approved grant proposal of the study with the editor.

1) The authors stated “Vagal afferent activation can thereby indirectly modulate key brain circuits involved in reward (Tellez et al., 2013) and energy homeostasis (see de Lartigue, 2016) as endogenous stimulation of the gut with nutrients evokes dopamine release in the dorsal striatum tracking caloric load (de Araujo et al., 2012; Tellez et al., 2016).” This is very weak, indirect and highly speculative.

We appreciate the reviewer’s concern about the cited research. However, we disagree with the assessment that these studies only provide weak and indirect support. The study by Tellez et al. (2013) showed that a gut lipid messenger (i.e., OEA) links dietary fat intake to dopamine levels. They write that “OEA anorectic effects are also known to be mediated by the vagus nerve (17). Indeed, intracerebral OEA infusions failed to influence dopamine release in either LF or HF mice (fig. S6).” Therefore, we consider this study strong and direct evidence. The study by Tellez et al. (2016) showed that optogenetic stimulation of the dorsal and ventral striatum can substitute for hedonic vs. energetic qualities of sugar intake, identifying the representation of “nutritional value” in the striatum. Although this study does not directly demonstrate a link to vagal afferent signals, this direct and strong evidence is provided by the paper of Han et al. (*Cell* 2018), which is cited in the sentence preceding the quoted section. Further crucial evidence on vagal afferent signals is reviewed in the cited work by G. de Lartigue (2016) and de Araujo, Ferreira, Tellez, Ren, and Yeckel (2012). Collectively, we do not see a good reason to discount this body of preclinical work as “highly speculative”, especially since most human

data provides only indirect evidence on neurotransmission in response to VNS.

2) Imaging studies investigating the effect of tVNS on the brain show no effect at all on the dorsal striatum (Dietrich et al., 2008; Frangos, Ellrich, & Komisaruk, 2015; Kraus et al., 2013; Yakunina et al., 2017) but tVNS activates the nucleus of the solitary tract, bilateral spinal trigeminal nucleus, dorsal raphe, locus coeruleus, and contralateral parabrachial area, amygdala, and nucleus accumbens. Accordingly, the following statement “human neuroimaging studies using fMRI have shown enhanced activity in.... striatum after concurrent taVNS (Frangos, Ellrich, & Komisaruk, 2015; Kraus et al., 2007)” is not accurate.

We appreciate having the opportunity to clarify the concern raised by the reviewer. First, we would like to point out that Frangos et al. (2015) report significant effects of taVNS throughout the striatum in the putamen, caudate (dorsal striatum), and nucleus accumbens (ventral striatum). As shown in Table 1 of that paper, they observed effects in the caudate, putamen and nucleus accumbens, albeit not significantly in all contrasts. Second, Kraus et al. (2007) reported activations that visibly extend into the dorsal striatum (i.e., the putamen) although this is not reported in the table. Third, the Yakunina et al. (2017) study was not cited in the previous version of our manuscript as support, although it reports increased activation in the caudate and putamen in the contrasts tragus and cymba conchae vs. sham. Moreover, the paper also reviews earlier studies as providing evidence regarding a modulation of the dorsal striatum by tVNS. We agree that the paper by Kraus et al. (2013) does not show a modulation of the dorsal striatum. Yet, this cannot be taken as conclusive evidence supporting the null hypothesis because the absence of an effect in some small studies could be simply due to thresholding results at a group level to avoid false positives. This leads to reduced power and the impression of a lower rate of replication across studies. Nevertheless, we are confident that there is enough evidence in the published literature to support our statement.

3) The authors assumed that the tVNS has an effect on dopamine, but there is no scientific proof of that. Instead VNS enhances the level of NA and GABA. There is plenty of evidence instead about the link iVNS augmented the release of NA in the LC and hippocampus (Dorr and Debonnel, 2006; Follesa et al., 2007; Groves et al., 2005; Hassert et al., 2004; Hulsey et al., 2017; Raedt et al., 2011; Roosevelt et al., 2006). Specifically, about tVNS salivary alpha-amylase (sAA) levels have been proposed to be an indirect marker for central NA system activation (Chatterton, Vogelsong, Lu, Ellman, & Hudgens, 1996; Warren, van den Brink, Nieuwenhuis, & Bosch, 2017). Two auricular tVNS studies have reported increased sAA levels in the active stimulation compared to sham condition (Ventura-Bort et al., 2018; Warren et al., 2019), which lends additional support to the assumption that tVNS modulates the NA system. About GABA a single-proton emission computed tomography study carried out in epileptic patients revealed that iVNS enhanced GABA receptor density in frontal and frontotemporal areas (Marrosu et al., 2003). Further, the short-interval intracortical inhibition,

a transcranial magnetic stimulation paradigm indicative of GABA-A activity, was substantially enhanced in right motor cortex after active tVNS (Capone et al., 2015).

If the authors had a priori hypothesis about dopamine why they did not measure dopamine levels via blood samples? Without any dopaminergic marker to sustain their hypothesis, the theoretical background proposed by the authors is very shaky.

We thank the reviewer for the well-researched comment on VNS-induced changes in neurotransmission. First, we agree with the reviewer that VNS affects transmitter systems other than dopamine (DA). Second, the reviewer cites previous work on VNS-induced modulation of the noradrenergic (NA) system including indirect evidence for central NA activation. However, it is important to keep in mind that such changes can be secondary to changes in dopamine because dopamine acts as a precursor to NA in the brain. For example, Landau et al. (2015) conducted a monoaminergic PET study during acute VNS in pigs and they discuss this issue in more detail:

“Interestingly, the results of this sample of 6 animals suggest that VNS exerts a significant inhibitory influence mostly in limbic, thalamic and cortical regions. The central distributions of monoamines are well documented with a prevalence of DA terminals in striatum while NA terminals predominate in cortical, limbic and thalamic regions. Despite the low density of DA innervation in non-striatal areas, it is however possible that part of the effect of VNS on yohimbine binding is mediated by release of DA as well as of NA. Indeed, electrical stimulation of the LC elicits both NA and DA release from NA terminals in rat cortex [32], [33] and many studies support crosstalk between the two ligands and their respective receptors [12].”

Third, whereas there is no proof that tVNS affects dopamine in humans yet, there is conclusive evidence that VNS does, in fact, alter dopamine transmission in animals (Han et al., 2018). Additional studies have shown that tVNS produces comparable results to VNS (He et al., 2013) and we are aware that animal studies are currently on the way to test whether tVNS produces comparable dopaminergic effect to invasive VNS. However, in order to conclusively test whether tVNS alters dopaminergic transmission in humans, a strong behavioral precedent is necessary because there is currently no way to directly measure dopamine transmission in humans without PET imaging. To get approval for a dopamine PET study, however, a strong rationale based on indirect evidence is usually required to justify the exposure of healthy participants to radiation elicited by the PET ligands.

Fourth, we can assure the reviewer that we would have assessed blood levels of dopamine if there was a way to quantify changes in dopamine neurotransmission based on peripheral blood markers. Whereas there are peripheral markers of dopaminergic drug interventions (e.g., levodopa or lactate), this cannot be used to investigate the effect of a brain stimulation technique that primarily acts on the CNS, to the best of our knowledge.

Lastly, we would like to point out that behavioral changes can also be useful to guide follow-up studies on neurotransmission. Invigoration has been previously linked to changes in dopamine transmission and our results are unlikely to be explained by an increase in GABAergic signaling. This approach

is characteristic for behavioral neuroscience and we feel that the value of detailed behavioral investigations such as provided by our study should not be discounted (Krakauer, Ghazanfar, Gomez-Marin, MacIver, & Poeppel, 2017). Nevertheless, we agree with the reviewer that we have perhaps put too much emphasis on a potential dopaminergic mechanism of taVNS and we are positive that the reviewer will appreciate reading a more balanced account of potential mechanisms in the revised version.

4) The authors stated: “Here, we tested whether non-invasive taVNS—applied to emulate interoceptive feedback signals emitted from the gut—would modulate instrumental behavior that is working for rewards (food or money).”

This prediction would make sense in the case of cervical tVNS (or iVNS), but not in the case of tVNS.

The use of auricular and cervical tVNS might have specific effects on the neuromodulation of cognitive processes because these tVNS approaches rely on the activation of only afferent (auricular tVNS) or both afferent and efferent fibers (cervical tVNS). Given that tVNS activates only afferent fibers it is not possible for this method to emulate interoceptive feedback signals emitted from the gut because this device does not mediate any effects coming from or getting to the gut.

We thank the reviewer for giving us the opportunity to clarify this statement. There is ample evidence demonstrating the crucial role of vagal afferents in forwarding signals related to energy intake from the gut to the brain (Cork, 2018; de Araujo, 2016; G. de Lartigue, 2016; Egerod et al., 2018; Grabauskas & Owyang, 2017; Yao et al., 2018). In contrast, efferent vagal signals from the brain to the gut do play a role as well, for example, in digestion effectively forming a feedback loop (Anselmi, Toti, Bove, & Travagli, 2017; Hong et al., 2019; Teckentrup et al., 2020). Precisely due to this circuit design, it is not necessary to a) stimulate efferent fibers to emulate interoceptive feedback signals emitted from the gut and to b) stimulate efferent fibers to cause efferent effects that are mediated by the NTS.

Thus, we disagree with the strong statement that this device does not mediate any effects coming from or getting to the gut. Illustratively, we have recently shown that acute taVNS leads to a reduction of gastric myoelectric frequency, a marker of gastric motility (Teckentrup et al., 2019). In this study, we applied a comparable single-blind randomized crossover taVNS stimulation protocol. Moreover, another study using taVNS during surgery found a similar reduction of gastric frequency (recorded via an invasive procedure (Hong et al., 2019)). Therefore, there is good evidence that taVNS can even elicit efferent vagal signals much like invasive VNS can do.

Bibliography

- Ainley, V., Apps, M. A., Fotopoulou, A., & Tsakiris, M. (2016). 'Bodily precision': a predictive coding account of individual differences in interoceptive accuracy. *Philos Trans R Soc Lond B Biol Sci*, *371*(1708). doi:10.1098/rstb.2016.0003
- Anselmi, L., Toti, L., Bove, C., & Travagli, R. A. (2017). Vagally mediated effects of brain stem dopamine on gastric tone and phasic contractions of the rat. *Am J Physiol Gastrointest Liver Physiol*, *313*(5), G434-G441. doi:10.1152/ajpgi.00180.2017
- Bühler, M., Vollstadt-Klein, S., Kobiella, A., Budde, H., Reed, L. J., Braus, D. F., . . . Smolka, M. N. (2010). Nicotine dependence is characterized by disordered reward processing in a network driving motivation. *Biol Psychiatry*, *67*(8), 745-752. doi:10.1016/j.biopsych.2009.10.029
- Chen, G., Saad, Z. S., Britton, J. C., Pine, D. S., & Cox, R. W. (2013). Linear mixed-effects modeling approach to fMRI group analysis. *Neuroimage*, *73*, 176-190. doi:10.1016/j.neuroimage.2013.01.047
- Chong, T. T., Bonnelle, V., Manohar, S., Veromann, K. R., Muhammed, K., Tofaris, G. K., . . . Husain, M. (2015). Dopamine enhances willingness to exert effort for reward in Parkinson's disease. *Cortex*, *69*, 40-46. doi:10.1016/j.cortex.2015.04.003
- Cork, S. C. (2018). The role of the vagus nerve in appetite control: Implications for the pathogenesis of obesity. *J Neuroendocrinol*, e12643. doi:10.1111/jne.12643
- de Araujo, I. E. (2016). High fat takes the low road to the brain's reinforcement system. *Current Opinion in Behavioral Sciences*, *9*, 158-162. doi:10.1016/j.cobeha.2016.04.013
- de Araujo, I. E., Ferreira, J. G., Tellez, L. A., Ren, X., & Yeckel, C. W. (2012). The gut-brain dopamine axis: a regulatory system for caloric intake. *Physiol Behav*, *106*(3), 394-399. doi:10.1016/j.physbeh.2012.02.026
- de Araujo, I. E., Schatzker, M., & Small, D. M. (2019). Rethinking Food Reward. *Annu Rev Psychol*. doi:10.1146/annurev-psych-122216-011643
- de Lartigue, G. (2016). Role of the vagus nerve in the development and treatment of diet-induced obesity. *J Physiol*, *594*(20), 5791-5815. doi:10.1113/JP271538
- de Lartigue, G. (2016). Role of the vagus nerve in the development and treatment of diet-induced obesity. *The Journal of Physiology*, *594*(20), 5791-5815. doi:10.1113/JP271538
- Dunn, B. D., Galton, H. C., Morgan, R., Evans, D., Oliver, C., Meyer, M., . . . Dalgleish, T. (2010). Listening to your heart. How interoception shapes emotion experience and intuitive decision making. *Psychol Sci*, *21*(12), 1835-1844. doi:10.1177/0956797610389191
- Egerod, K. L., Petersen, N., Timshel, P. N., Rekling, J. C., Wang, Y., Liu, Q., . . . Gautron, L. (2018). Profiling of G protein-coupled receptors in vagal afferents reveals novel gut-to-brain sensing mechanisms. *Mol Metab*, *12*, 62-75. doi:10.1016/j.molmet.2018.03.016
- Fallgatter, A. J., Neuhauser, B., Herrmann, M. J., Ehlis, A. C., Wagerer, A., Scheuerpflug, P., . . . Riederer, P. (2003). Far field potentials from the brain

- stem after transcutaneous vagus nerve stimulation. *J Neural Transm (Vienna)*, 110(12), 1437-1443. doi:10.1007/s00702-003-0087-6
- Frangos, E., Ellrich, J., & Komisaruk, B. R. (2015). Non-invasive Access to the Vagus Nerve Central Projections via Electrical Stimulation of the External Ear: fMRI Evidence in Humans. *Brain Stimul*, 8(3), 624-636. doi:10.1016/j.brs.2014.11.018
- Frangos, E., & Komisaruk, B. R. (2017). Access to Vagal Projections via Cutaneous Electrical Stimulation of the Neck: fMRI Evidence in Healthy Humans. *Brain Stimul*, 10(1), 19-27. doi:10.1016/j.brs.2016.10.008
- Grabauskas, G., & Owyang, C. (2017). Plasticity of vagal afferent signaling in the gut. *Medicina (Kaunas)*, 53(2), 73-84. doi:10.1016/j.medic.2017.03.002
- Han, W., Tellez, L. A., Perkins, M. H., Perez, I. O., Qu, T., Ferreira, J., . . . de Araujo, I. E. (2018). A Neural Circuit for Gut-Induced Reward. *Cell*, 175(3), 665-678 e623. doi:10.1016/j.cell.2018.08.049
- He, W., Jing, X. H., Zhu, B., Zhu, X. L., Li, L., Bai, W. Z., & Ben, H. (2013). The auriculo-vagal afferent pathway and its role in seizure suppression in rats. *BMC Neurosci*, 14, 85. doi:10.1186/1471-2202-14-85
- Herbert, B. M., Ulbrich, P., & Schandry, R. (2007). Interoceptive sensitivity and physical effort: implications for the self-control of physical load in everyday life. *Psychophysiology*, 44(2), 194-202. doi:10.1111/j.1469-8986.2007.00493.x
- Hong, G. S., Pinteá, B., Lingohr, P., Coch, C., Randau, T., Schaefer, N., . . . Pantelis, D. (2019). Effect of transcutaneous vagus nerve stimulation on muscle activity in the gastrointestinal tract (transVaGa): a prospective clinical trial. *Int J Colorectal Dis*, 34(3), 417-422. doi:10.1007/s00384-018-3204-6
- Krakauer, J. W., Ghazanfar, A. A., Gomez-Marin, A., MacIver, M. A., & Poeppel, D. (2017). Neuroscience Needs Behavior: Correcting a Reductionist Bias. *Neuron*, 93(3), 480-490. doi:10.1016/j.neuron.2016.12.041
- Kraus, T., Hosl, K., Kiess, O., Schanze, A., Kornhuber, J., & Forster, C. (2007). BOLD fMRI deactivation of limbic and temporal brain structures and mood enhancing effect by transcutaneous vagus nerve stimulation. *J Neural Transm (Vienna)*, 114(11), 1485-1493. doi:10.1007/s00702-007-0755-z
- Kraus, T., Kiess, O., Hosl, K., Terekhin, P., Kornhuber, J., & Forster, C. (2013). CNS BOLD fMRI effects of sham-controlled transcutaneous electrical nerve stimulation in the left outer auditory canal - a pilot study. *Brain Stimul*, 6(5), 798-804. doi:10.1016/j.brs.2013.01.011
- Kroemer, N. B., Guevara, A., Ciocanea Teodorescu, I., Wuttig, F., Kobiella, A., & Smolka, M. N. (2014). Balancing reward and work: anticipatory brain activation in NAcc and VTA predict effort differentially. *Neuroimage*, 102 Pt 2, 510-519. doi:10.1016/j.neuroimage.2014.07.060
- Kühnel, A., Teckentrup, V., Neuser, M. P., Huys, Q. J., Burrasch, C., Walter, M., & Kroemer, N. B. (2019). Stimulation of the vagus nerve reduces learning in a go/no-go reinforcement learning task. *bioRxiv*, 535260.
- Le Heron, C., Plant, O., Manohar, S., Ang, Y. S., Jackson, M., Lennox, G., . . . Husain, M. (2018). Distinct effects of apathy and dopamine on effort-based decision-making in Parkinson's disease. *Brain*. doi:10.1093/brain/awy110
- Mazzoni, P., Hristova, A., & Krakauer, J. W. (2007). Why don't we move faster? Parkinson's disease, movement vigor, and implicit motivation. *J Neurosci*, 27(27), 7105-7116. doi:10.1523/JNEUROSCI.0264-07.2007

- McGinty, V. B., Lardeux, S., Taha, S. A., Kim, J. J., & Nicola, S. M. (2013). Invigoration of reward seeking by cue and proximity encoding in the nucleus accumbens. *Neuron*, *78*(5), 910-922. doi:10.1016/j.neuron.2013.04.010
- Panigrahi, B., Martin, K. A., Li, Y., Graves, A. R., Vollmer, A., Olson, L., . . . Dudman, J. T. (2015). Dopamine Is Required for the Neural Representation and Control of Movement Vigor. *Cell*, *162*(6), 1418-1430. doi:10.1016/j.cell.2015.08.014
- Raudenbush, S. W., & Bryk, A. S. (2002). *Hierarchical linear models: Applications and data analysis methods* (Vol. 1): Sage.
- Sclocco, R., Garcia, R. G., Kettner, N. W., Isenburg, K., Fisher, H. P., Hubbard, C. S., . . . Napadow, V. (2019). The influence of respiration on brainstem and cardiovagal response to auricular vagus nerve stimulation: A multimodal ultrahigh-field (7T) fMRI study. *Brain Stimul*, *12*(4), 911-921. doi:10.1016/j.brs.2019.02.003
- Suarez, A. N., Hsu, T. M., Liu, C. M., Noble, E. E., Cortella, A. M., Nakamoto, E. M., . . . Kanoski, S. E. (2018). Gut vagal sensory signaling regulates hippocampus function through multi-order pathways. *Nature Communications*, *9*(1), 1-15.
- Teckentrup, V., Neubert, S., Santiago, J. C., Hallschmid, M., Walter, M., & Kroemer, N. B. (2020). Non-invasive stimulation of vagal afferents reduces gastric frequency. *Brain stimulation*, *13*(2), 470-473.
- Tellez, L. A., Han, W., Zhang, X., Ferreira, T. L., Perez, I. O., Shammah-Lagnado, S. J., . . . de Araujo, I. E. (2016). Separate circuitries encode the hedonic and nutritional values of sugar. *Nat Neurosci*, *19*(3), 465-470. doi:10.1038/nn.4224
- Tellez, L. A., Medina, S., Han, W., Ferreira, J. G., Licona-Limon, P., Ren, X., . . . de Araujo, I. E. (2013). A gut lipid messenger links excess dietary fat to dopamine deficiency. *Science*, *341*(6147), 800-802. doi:10.1126/science.1239275
- Yakunina, N., Kim, S. S., & Nam, E. C. (2017). Optimization of Transcutaneous Vagus Nerve Stimulation Using Functional MRI. *Neuromodulation*, *20*(3), 290-300. doi:10.1111/ner.12541
- Yao, G., Kang, L., Li, J., Long, Y., Wei, H., Ferreira, C. A., . . . Wang, X. (2018). Effective weight control via an implanted self-powered vagus nerve stimulation device. *Nat Commun*, *9*(1), 5349. doi:10.1038/s41467-018-07764-z

***REVIEWERS' COMMENTS:

Reviewer #1 (Remarks to the Author):

The authors have done a very good job in revising the manuscript and alleviated my concerns on an interesting manuscript.

Reviewer # 2 (Remarks to the Author)

[blank; see SPECIFIC EDITORIAL REQUESTS, ABOVE]

Reviewer #3 (Remarks to the Author):

I believe the authors have appropriately addressed my concerns upon providing additional statistical analyses as well as by providing clarifications on how they interpreted any potential effects of sham stimulation. I also consider that their premises linking the vagus nerve to reward systems in brain are well grounded on currently available animal and human data.

Response letter

We thank the reviewers for their appraisal of our revised manuscript. Based on the feedback provided by Reviewer 2 and the editors, we have revised the manuscript accordingly.

Reviewer #1 (Remarks to the Author):

The authors have done a very good job in revising the manuscript and alleviated my concerns on an interesting manuscript.

We thank the reviewer for his or her assessment of the revision and the very constructive feedback in the first round.

Reviewer #2

[in the remarks to the editors]

1. Reviewer 2 has requested that your arguments in response to their suggestion of purely motor effects be integrated into the manuscript.

We thank the reviewer for his or her suggestion that the editors endorsed and gladly incorporated this aspect into the revised discussion:

“The associations of invigoration slopes with reward magnitude and rated wanting, but not rated exhaustion in our task support the interpretation that the speed of invigoration is primarily related to the prospective benefit of actions and largely independent of the costs incurred by effort. These effects cannot be explained by a non-motivational motor effect. First, there is good evidence that invigoration is dependent on reward magnitude [17] and we also report a quicker invigoration when large rewards are at stake in our task. Second, invigoration is associated with wanting ratings, but not ratings of exertion on the same trial (Fig. 2e) or exertion on the previous trial. Relatedly, motivational aspects of motor control contribute to motor symptoms in Parkinson’s disease [41] which is modulated by alterations in dopaminergic transmission [42]. Thus, one plausible explanation is that the hypothesized taVNS-induced increases in dopamine tone act comparable to an increase in the average rate of rewards [31, 43, 44]. (p. 9 f)

2. The reviewer also suggests that the abstract clearly state that this study explored acute taVNS, and that the effect was on the implicit invigoration of tapping measure of motivation, but not explicit self report measures. The editorial team would strongly encourage you to make these changes.

We thank the reviewer for pointing this clarification concerning the scope of our work in the abstract. We have incorporated the suggested changes and are positive that this provides a fair and balanced account of the results.

“Abstract

Interoceptive feedback transmitted via the vagus nerve plays a vital role in motivation by tuning actions according to physiological needs. Whereas vagus nerve stimulation (VNS) reinforces actions in animals, motivational effects elicited by VNS in humans are still largely elusive. Here, we applied non-invasive transcutaneous auricular VNS (taVNS) on the left or right ear while participants exerted effort to earn rewards using a randomized cross-over design (vs. sham). In line with preclinical studies, acute taVNS enhances invigoration of effort, and stimulation on the left side primarily facilitates invigoration for food rewards. In contrast, we do not find conclusive evidence that acute taVNS affects effort maintenance or wanting ratings. Collectively, our results suggest that taVNS enhances reward-seeking by boosting invigoration, not effort maintenance and that the stimulation side affects generalization beyond food reward. Thus, taVNS may enhance the pursuit of prospective rewards which may pave avenues to treat motivational deficiencies.” (p. 2)

Reviewer #3 (Remarks to the Author):

I believe the authors have appropriately addressed my concerns upon providing additional statistical analyses as well as by providing clarifications on how they interpreted any potential effects of sham stimulation. I also consider that their premises linking the vagus nerve to reward systems in brain are well grounded on currently available animal and human data.

We thank the reviewer for his or her feedback on the revision and appreciated the thoughtful input that we gladly incorporated.

Reviewer #4

[blank]